EMBO
reports

# 1,10-phenanthroline inhibits sumoylation and reveals that yeast SUMO modifications are highly transient

J Bryan McNeil[1], Su-Kyong Lee[1], Anna Oliinyk[1], Sehaj Raina[1], Jyoti Garg[2], Marjan Moallem[1], Verne Urquhart-Cox[1], Jeffrey Fillingham[2], Peter Cheung[1] & Emanuel Rosonina[1✉]

## Abstract

**The steady-state levels of protein sumoylation depend on relative rates of conjugation and desumoylation. Whether SUMO modifications are generally long-lasting or short-lived is unknown. Here we show that treating budding yeast cultures with 1,10-phenanthroline abolishes most SUMO conjugations within one minute, without impacting ubiquitination, an analogous post-translational modification. 1,10-phenanthroline inhibits the formation of the E1-SUMO thioester intermediate, demonstrating that it targets the first step in the sumoylation pathway. SUMO conjugations are retained after treatment with 1,10-phenanthroline in yeast that express a defective form of the desumoylase Ulp1, indicating that Ulp1 is responsible for eliminating existing SUMO modifications almost instantly when de novo sumoylation is inhibited. This reveals that SUMO modifications are normally extremely transient because of continuous desumoylation by Ulp1. Supporting our findings, we demonstrate that sumoylation of two specific targets, Sko1 and Tfg1, virtually disappears within one minute of impairing de novo sumoylation. Altogether, we have identified an extremely rapid and potent inhibitor of sumoylation, and our work reveals that SUMO modifications are remarkably short-lived.**

**Keywords** 1,10-Phenanthroline; Desumoylation; SUMO; Sumoylation; Ulp1
**Subject Category** Post-translational Modifications & Proteolysis

## Introduction

Many eukaryotic proteins, particularly those that associate with chromatin, undergo SUMO post-translational modification (PTM), which can affect their stability, activity, molecular interactions, or localization (Flotho and Melchior, 2013; Chymkowitch et al, 2015; Rosonina et al, 2017; Baig et al, 2021; Zhao, 2018; Boulanger et al, 2021). Sumoylation occurs through a cascade of enzymatic activities, starting with activation by the dimeric E1 enzyme (Olsen et al, 2010; Flotho and Melchior, 2013). During activation, which requires ATP and magnesium, the SUMO peptide is first adenylated, then transferred to a Cys residue within the Uba2 subunit of E1, through a thioester bond, and AMP is released. SUMO is then transferred to the E2 conjugating enzyme, Ubc9, again through a Cys-mediated thioester linkage. Often facilitated by E3 ligases, E2 then catalyzes attachment of SUMO to specific Lys residues within target proteins through isopeptide bonds. Although the process of sumoylation is analogous to ubiquitination, there are far fewer members of the SUMO enzyme classes, including single E1 and E2 enzymes in both budding yeast and humans, suggesting that the sumoylation levels of hundreds of SUMO targets can be modulated by regulating just one enzyme (Flotho and Melchior, 2013). Illustrating the high level of conservation of sumoylation across eukaryotes, human UBC9 can functionally replace yeast Ubc9, and one of the three main human SUMO isoforms, SUMO1, can complement yeast lacking *SMT3*, which encodes its sole SUMO peptide (Hamza et al, 2015; Newman et al, 2017). SUMO modifications can be reversed through the action of SUMO proteases (desumoylases), including members of the SENP family in mammals and Ulp1 and Ulp2 in yeast (Hickey et al, 2012; Li and Hochstrasser, 2000, 1999). Steady-state levels of SUMO conjugation, therefore, depend on the relative activities of sumoylation and desumoylation machineries. However, whether SUMO conjugation events are generally long-lasting or transient is not known. Here, we report that 1,10-phenanthroline, a drug often used to inhibit transcription, is an extremely potent and rapid inhibitor of sumoylation in yeast. Using this drug, we demonstrate that most SUMO modifications are remarkably transient, primarily because of continuous desumoylation by Ulp1.

## Results and discussion

### 1,10-phenanthroline is a potent inhibitor of sumoylation

Transcription-related proteins comprise one of the largest groups of SUMO targets identified across eukaryotic species (Makhnevych et al, 2009; Albuquerque et al, 2015; Esteras et al, 2017; Hendriks et al, 2017; Rosonina et al, 2017). To determine whether SUMO conjugation depends on ongoing transcription, we inhibited transcription in budding yeast by different methods and assessed overall sumoylation levels. Lysates were prepared from cultures of the *rpb1-1* strain (Nonet et al, 1987), which carries a temperature-sensitive mutation in the

[1]Department of Biology, York University, 4700 Keele Street, Toronto, ON M3J 1P3, Canada. [2]Department of Chemistry and Biology, Toronto Metropolitan University, 350 Victoria Street, Toronto, ON M5B 2K3, Canada. ✉E-mail: rosonina@yorku.ca

Rpb1 subunit of RNA polymerase II (RNAPII), or from the wild-type strain W303a treated with thiolutin or 1,10-phenanthroline. Both of these drugs have been widely used as transcriptional inhibitors in yeast, and although their modes of action are not fully known, they are thought to impair transcription through different mechanisms (Grigull et al, 2004; preprint: Qiu et al, 2021; Eshleman et al, 2020). As part of the transition process between transcription initiation and elongation, the C-terminal domain (CTD) repeats of RNAPII become highly phosphorylated on Ser-2 residues (S2p) (Bowman and Kelly, 2014). Consistent with impaired transcription, lysates from the *rpb1-1* strain grown at the non-permissive temperature (37 °C) and from W303a treated with 1,10-phenanthroline both showed reduced S2p levels compared to their respective controls (Fig. 1A). Although S2p levels were not reduced in lysates from thiolutin-treated cultures, we confirmed by RT-qPCR that this treatment inhibits transcription (see Fig. 1E). Overall sumoylation levels were then examined by SUMO immunoblot of the lysates. Whereas transcription inhibition by inactivation of Rpb1 (in the *rpb1-1* strain at 37 °C) or by treatment with thiolutin did not diminish sumoylation, remarkably, treatment of cultures with 1,10-phenanthroline nearly abolished all SUMO conjugations (Fig. 1A). Treatment of cells with the structural analog 1,7-phenanthroline did not impact sumoylation levels, indicating that the effect is specific to the 1,10 isomer (Fig. 1A). These data demonstrate that, whereas impaired transcription itself does not reduce overall sumoylation levels, unexpectedly, 1,10-phenanthroline is an extremely potent inhibitor of sumoylation.

1,10-phenanthroline is a chelator of zinc and other heavy metals, but it has been proposed to function in transcription inhibition by sequestering magnesium (Kiss, 1994; Grigull et al, 2004). To explore if reduced sumoylation by 1,10-phenanthroline is part of a general inhibition of metal-dependent cellular processes, we tested whether the drug impacts overall levels of ubiquitination, which is highly analogous to sumoylation, requiring the same types of enzymatic reactions (Flotho and Melchior, 2013; Rape, 2018). Cultures of yeast expressing HA epitope-tagged ubiquitin were treated with 1,10-phenanthroline, and as shown in Fig. 1B, ubiquitin conjugation levels were not impacted while sumoylation levels were dramatically reduced, implying that 1,10-phenanthroline impairs sumoylation specifically. Further characterizing the effect, we found that inhibition of sumoylation by 1,10-phenanthroline is dose-dependent with SUMO conjugation levels somewhat reduced at 30 μg/mL (0.17 mM) and virtually abolished at 300–500 μg/mL (1.7–2.8 mM; Fig. 1C). For comparison, the drug has been used at 100–500 μg/mL to inhibit transcription (Lewicki et al, 2015; Grigull et al, 2004; Martin et al, 2021). Dramatically, in a time course analysis, we found that 1,10-phenanthroline works extremely rapidly, with most SUMO conjugations disappearing within 1 min of treatment (Fig. 1D). To compare the rate of inhibition of sumoylation and transcription by 1,10-phenanthroline, RNA was isolated from cultures treated with the drug for different durations, then RT-qPCR was performed using PCR primers that span intron-exon junctions for two ribosomal protein genes (RPGs). Since RPG RNA splicing in yeast is very rapid, the level of intron-containing transcripts present at any time can be an approximate measure of nascent transcription (Wallace and Beggs, 2017). As shown in Fig. 1E, whereas 1,10-phenanthroline reduced transcription of the tested genes to ~55% after 1 min and to ~35% at 30 min, the drug eliminated nearly 95% of SUMO conjugations within just the first minute. These data demonstrate that 1,10-phenanthroline is a remarkably rapid and potent inhibitor of sumoylation in yeast and

that it acts significantly faster and more effectively as a sumoylation inhibitor than a transcription inhibitor.

## 1,10-phenanthroline blocks the first step of the sumoylation pathway

To explore possible mechanisms by which 1,10-phenanthroline reduces SUMO conjugation levels, we examined whether the drug impacts in vitro sumoylation reactions using purified components of the SUMO pathway. In vitro sumoylation was carried out using recombinant human E1 (SAE1/UBA2 heterodimer), UBC9, the SUMO1 or SUMO2 isoform, and as a sample substrate, RanGAP1. DMSO or three different concentrations of 1,10-phenanthroline or 1,7-phenanthroline were included in the reactions. Strongly supporting our in vivo analysis, the addition of 1,10-phenanthroline, particularly at 20 mM (3.6 mg/mL), greatly diminished the production of sumoylated RanGAP1, whereas 1,7-phenanthroline had no effect (Figs. 2A and S1A). The inhibition of sumoylation is not likely the result of magnesium chelation by 1,10-phenanthroline since the addition of an excess of MgCl$_2$ (30 mM MgCl$_2$ versus 20 mM 1,10-phenanthroline) did not reverse the inhibition in vitro (Fig. 2B), and because reducing magnesium levels in culture medium by tenfold did not enhance the effects of the drug in vivo (Appendix Fig. S1B). In addition, treatment of cultures with the commonly used zinc chelating agent TPEN had no impact on sumoylation levels, arguing that SUMO conjugation is not particularly sensitive to reduced zinc availability (Kawamata et al, 2017; Fig. S1C). This demonstrates that 1,10-phenanthroline inhibits SUMO conjugation not only in living yeast but also using purified human components of the pathway through a mechanism that does not involve the chelation of magnesium or zinc.

SUMO immunoblots of in vitro sumoylation reactions also revealed a second SUMO2-containing species, which derives from sumoylated UBC9 (Fig. 2A). Although UBC9 itself can become modified by SUMO through an isopeptide bond at Lys 14 (Knipscheer et al, 2008), treatment of reactions with reducing agent (DTT) prior to electrophoresis nearly abolished the UBC9-derived signal in the immunoblots, indicating that it derives largely from the thioester bond-connected UBC9 ~ SUMO intermediate (Fig. 2C). Intriguingly, the addition of 1,10-phenanthroline strongly impaired formation of the UBC9 ~ SUMO species regardless of whether the RanGAP1 substrate was present (Fig. 2A,D) or omitted (Fig. 2D). These data indicate that 1,10-phenanthroline acts to inhibit an early step in the sumoylation pathway, prior to conjugation of substrates. More specifically, 1,10-phenanthroline inhibits the formation of the UBC9 ~ SUMO thioester intermediate either directly or by impairing the prior step, formation of the E1~SUMO intermediate, which is catalyzed by the E1 enzyme itself.

To distinguish between these possibilities, we examined whether treatment with 1,10-phenanthroline prevents the thioester linkage of SUMO to Ubc9 and/or to the E1 subunit Uba2 in living yeast. Cultures of strains expressing HA-tagged forms of Ubc9 or Uba2, or an isogenic untagged strain, were treated with DMSO, 1,10-phenanthroline, or thiolutin, and lysates were prepared and analyzed by HA immunoblot in either the absence or presence of DTT. Higher molecular-weight forms of Ubc9 and Uba2 were readily observed in the DMSO or thiolutin-treated samples, and these forms were greatly reduced when DTT was present, indicating that they represent thioester-linked SUMO~Ubc9 and SUMO~Uba2, respectively (red arrowheads in Fig. 2E). Intriguingly, treatment of cultures with 1,10-phenanthroline

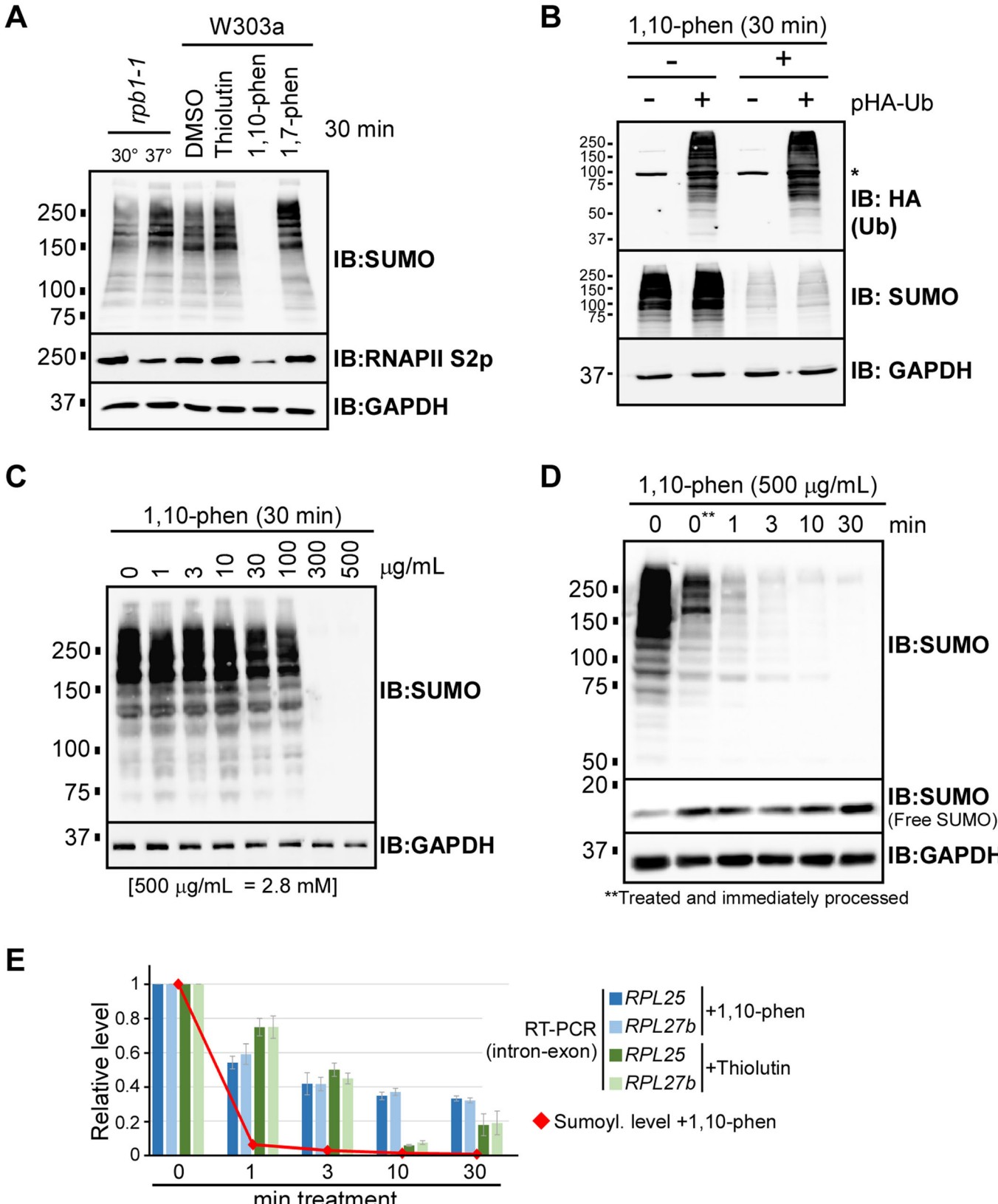

**Figure 1. 1,10-phenanthroline is a potent inhibitor of sumoylation in yeast.**

(A) Cultures of the *rpb1-1* or W303a yeast strains were treated as indicated (see "Methods"), then lysates were prepared and analyzed by SUMO, RNAPII S2p, and GAPDH immunoblots (IBs). The RNAPII S2p antibody recognizes repeats of the C-terminal domain (CTD) of Rpb1 subunit of RNAPII that are phosphorylated on Ser-2 residues, which occurs during transcription elongation. GAPDH serves as a loading control. *Phenanthroline* is abbreviated as *phen* in figures. (B) The W303a strain was transformed with a plasmid that expresses the ubiquitin peptide (Ubi4) with an N-terminal HA-tag. This strain, or its parent lacking the plasmid, were grown and treated with 500 μg/mL of 1,10-phenanthroline, or the same volume of DMSO, and lysates were prepared and analyzed by HA (i.e., ubiquitin), SUMO, and GAPDH immunoblots. Asterisk (*) refers to a non-specific band on the HA blot. (C) Cultures of the W303a strain were grown and treated with the indicated concentrations of 1,10-phenanthroline for 30 min. Lysates were prepared and analyzed by SUMO and GAPDH immunoblots. (D) Cultures of the W303a strain were grown and treated with 500 μg/mL (~2.8 mM) of 1,10-phenanthroline for the indicated durations, then lysates were prepared and analyzed by SUMO, RNAPII S2p, and GAPDH immunoblots. A blot showing unconjugated SUMO peptides ("Free SUMO") is included. The sample marked 0** was treated with 1,10-phenanthroline, then processed immediately. (E) Comparison of the rate of inhibition of transcription and SUMO conjugation by 1,10-phenanthroline. RNA was isolated from W303a yeast cultures that had been treated with 500 μg/mL of 1,10-phenanthroline (or 4 μg/mL of thiolutin, for comparison) for the indicated times, then RT-qPCR was performed on the indicated genes, using primers that span intron-exon junctions, as an approximation of nascent transcription levels. Averages of three replicates are plotted relative to the zero time-point, with standard deviations indicated as error bars. For comparison, levels of overall SUMO conjugation at the corresponding time points after treatment with the drug are also plotted, as determined by densitometry of the immunoblot shown in (D). Source data are available online for this figure.

dramatically reduced the formation of thioester-linked Ubc9 and Uba2. Since the formation of Uba2~SUMO occurs prior to the formation of Ubc9~SUMO, we therefore conclude that 1,10-phenanthroline inhibits the activity of the SUMO E1 enzyme.

Considering that 1,10-phenanthroline inhibits sumoylation in a human in vitro system (as described above), we tested whether treatment of cultured human cells would also lead to reduced sumoylation. Soluble protein lysates were prepared from 293T and HeLa cells treated with 1,10-phenanthroline for different durations. SUMO1 and SUMO2/3 immunoblot analysis indicated that the drug effectively reduced all forms of sumoylation by 30 min, but generally not before 5 min (Fig. S1D). Vegetative *Tetrahymena* harbor only a small number of detectable SUMO conjugates (Nasir et al, 2015), and treatment with 1,10-phenanthroline eliminated sumoylation of its major conjugate, also between 5 and 30 min (Fig. S1E). Therefore, although the drug appears to be a general inhibitor of sumoylation across eukaryotic species and in vitro, treatment with 1,10-phenanthroline abolishes SUMO conjugation significantly faster in yeast than in other organisms.

## SUMO conjugation is remarkably transient in yeast because of continuous desumoylation by Ulp1

Our finding that most sumoylation events disappear within 1 min of impairing de novo SUMO conjugation by treatment with 1,10-phenanthroline indicates that SUMO modifications are surprisingly extremely transient. We explored whether either of the two yeast SUMO proteases, Ulp1 or Ulp2, is responsible for continuous desumoylation by examining whether the near-total disappearance of SUMO conjugates after treatment with 1,10-phenanthroline is impacted by the absence of Ulp2 (in the *ulp2Δ* strain), whose gene is not essential, or by a mutation (*ulp1-mt*) that partially inactivates Ulp1 (Soustelle et al, 2004). The corresponding isogenic wild-type strains *ULP2* and *ULP1*, which correspond to background lab yeast strains BY4741 and MH1006, respectively, both showed a dramatic reduction in sumoylation levels after treatment with 1,10-phenanthroline, indicating that the effect is not specific to the W303a strain used for the experiments shown in Fig. 1 (Fig. 3A). Levels of unconjugated ("free") SUMO increased in both strains after addition of the drug, supporting the idea that desumoylation is responsible for the loss of SUMO conjugation ("Free SUMO" in Fig. 3A, and see Fig. 1D). Consistent with this, although deletion of *ULP2* showed a slight reversal of the effects of the drug, impairment

of Ulp1 activity completely reversed the disappearance of SUMO conjugates by 1,10-phenanthroline (Fig. 3A). Remarkably, these data imply that most sumoylation events in yeast are highly transient because of continuous desumoylation by Ulp1, at least under normal (i.e., optimal) growth conditions. To rule out the possibility that 1,10-phenanthroline reduces SUMO conjugation levels by elevating Ulp1 activity, we treated yeast lysates with recombinant Ulp1 and found that the drug did not enhance its desumoylation activity. Rather, it was somewhat decreased with higher drug concentrations (Fig. S1F). Together, these data strongly support the notion that when yeast are treated with 1,10-phenanthroline, E1 activity is rapidly inhibited, and existing SUMO modifications are rapidly eliminated by Ulp1.

To explore the transient nature of SUMO conjugation in a different way, we used the Anchor-away system, using a strain (Ubc9-AA) in which Ubc9 can be rapidly depleted from the nucleus by induced translocation to the cytoplasm upon treatment with rapamycin (Haruki et al, 2008; Moallem et al, 2023). Although sumoylation levels in the Ubc9-AA strain are constitutively very low (Moallem et al, 2023), nuclear depletion of Ubc9 causes a significant further decrease in overall sumoylation in this system, likely because the vast majority of SUMO-conjugated proteins are associated with chromatin, and therefore depend on nuclear Ubc9 for their modification (Baig et al, 2021). Strongly consistent with sumoylation being highly transient, we found that treatment with rapamycin for just 5 min was sufficient to eliminate most SUMO conjugations in the Ubc9-AA strain but not in its parental untagged strain (Figs. 3B and S1G). The loss of sumoylated proteins was not due to proteasome-mediated proteolysis of SUMO conjugates, since prior treatment of the strain with the proteasome inhibitor MG132 did not impact the drop in sumoylation (Fig. 3C), nor was it due to desumoylation by Ulp2 since the effect was also observed in its absence (Fig. 3D). Although, for unknown reasons, we were not able to generate a viable *ulp1-mt* version of the Ubc9-AA strain, having ruled out proteolysis and desumoylation by Ulp2, we believe that Ulp1 is likely responsible for the rapid loss of SUMO conjugations upon nuclear depletion of Ubc9. Supporting this, we found that the *ulp1-mt* mutation largely restored sumoylation levels in a strain with dramatically reduced Ubc9 function due to the *ubc9-6* mutation (Fig. 3E) (Baig et al, 2021). In all, these data are strongly consistent with a role for Ulp1 in continuously reversing most SUMO conjugation events. Although the *ulp1-mt* mutation counters the effects of 1,10-phenanthroline in reducing SUMO conjugation levels, it did not reverse lethality caused by exposure of yeast cultures to the drug, as

**A**

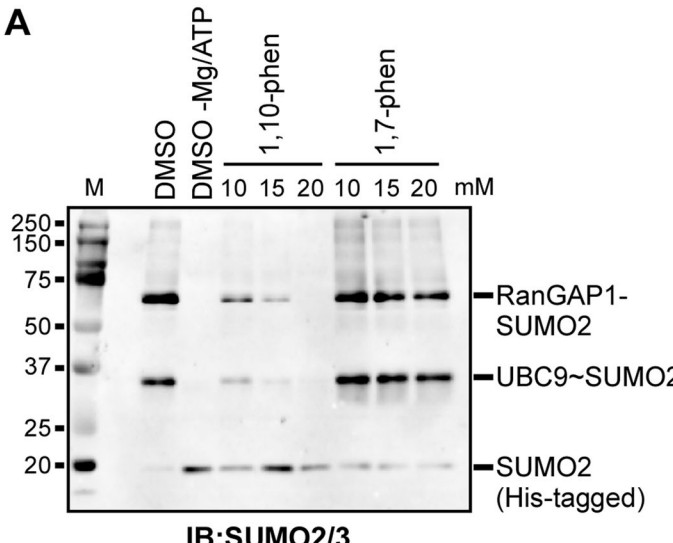

IB:SUMO2/3

**B**

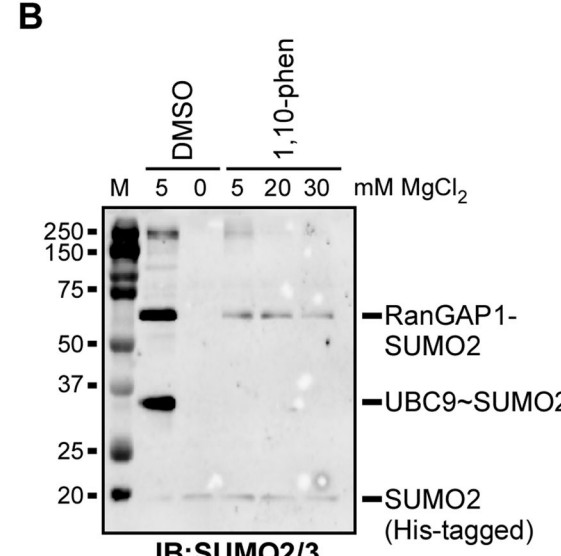

IB:SUMO2/3

**C**

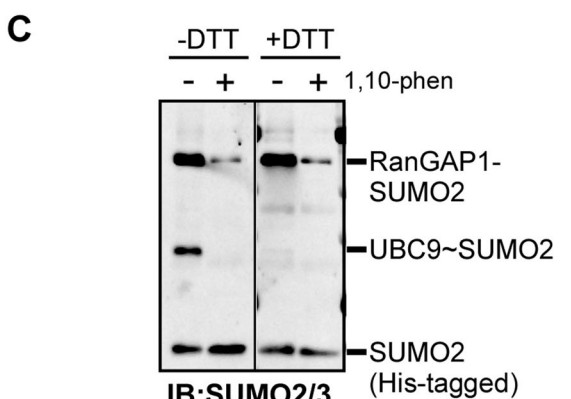

IB:SUMO2/3

**D**

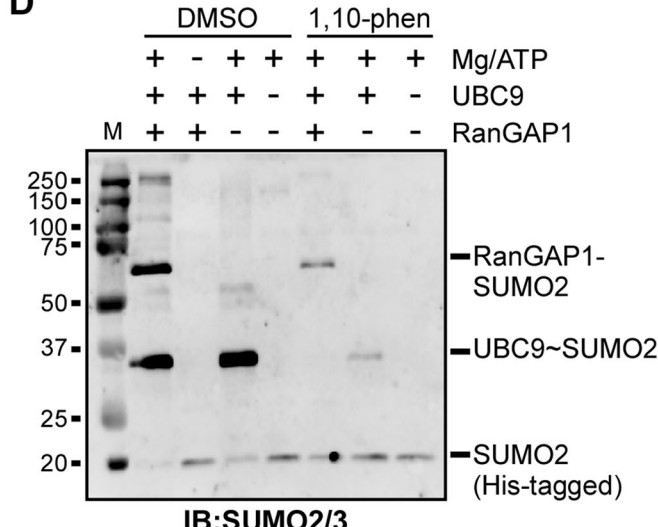

IB:SUMO2/3

**E**

IB:HA

**Figure 2. 1,10-phenanthroline inhibits an early step in the sumoylation pathway.**

(A) In vitro sumoylation reactions were carried out using RanGAP1 as a substrate and treated with DMSO (control) or the indicated concentrations of 1,10-phenanthroline or its structural isomer, 1,7-phenanthroline (see "Methods"). Reactions were analyzed by immunoblot with an antibody that recognizes SUMO2- and SUMO3-conjugated proteins (SUMO2/3). The sample labeled "DMSO -Mg/ATP" is a negative control and was carried out in the absence of MgCl$_2$ and ATP (Mg/ATP). M, protein size marker, with molecular masses indicated in kDa. (B) In vitro sumoylation reactions were carried out with DMSO or 20 mM (3.6 mg/mL) of 1,10-phenanthroline and the indicated concentrations of MgCl$_2$, then analyzed by SUMO2/3 immunoblot. Normally, 5 mM of MgCl$_2$ is included in the in vitro reactions. (C) In vitro sumoylation reactions were carried out in the presence or absence of 20 mM 1,10-phenanthroline, then 10 mM DTT was added to half of each reaction prior to electrophoresis. SUMO2/3 immunoblot of the samples is shown. (D) In vitro sumoylation reactions were carried in the presence of DMSO or 20 mM 1,10-phenanthroline with all reaction components, or lacking Mg/ATP, RanGAP1, or UBC9, as indicated. SUMO2/3 immunoblot of the reactions is shown. (E) Strains expressing HA-tagged Uba2 or Ubc9, or their isogenic untagged strains ("No tag"), were grown in culture and treated with 1,10-phenanthroline (500 μg/mL), thiolutin (4 μg/mL), or DMSO for 30 min. Lysates were then prepared (using IP buffer lacking DTT) and analyzed by immunoblot in the absence or presence of 10 mM DTT in the sample buffer (lacking 2-mercaptoethanol). Red arrowheads indicate the position of thioester SUMO-linked Ubc9 and Uba2. Non-specific bands detected by the HA antibody are indicated with asterisks (*). A shorter exposure of the Uba2-HA blots is shown at the bottom. Source data are available online for this figure.

shown in the growth curves in Fig. 3F. This suggests that the toxicity of 1,10-phenanthroline is due to effects of the drug that are not directly or immediately related to reduced SUMO conjugation levels. This is consistent with our recent finding that yeast cells can tolerate a dramatic reduction in sumoylation levels with little to no effect on viability (Moallem et al, 2023).

## 1,10-phenanthroline treatment reveals that sumoylation of Sko1 and Tfg1 is highly transient

Next, we examined the stability of the sumoylation of two transcription-related proteins that we have previously studied in detail, Sko1 and the large subunit of general transcription factor TFIIF, Tfg1 (Sri Theivakadadcham et al, 2019; Baig et al, 2021). Lysates prepared from a strain expressing HA-tagged Sko1 (Sko1-HA) were treated with 1,10-phenanthroline for different durations, then HA immunoprecipitation (IP) was followed by SUMO and HA immunoblotting. Sumoylated forms of IPed Sko1-HA can be detected in the SUMO immunoblot, but as we previously demonstrated, mono-sumoylated Sko1-HA can also be detected on HA blots as a slower migrating form the protein (red arrowhead in Fig. 4A). Consistent with the rapid disappearance of overall SUMO conjugations, sumoylated forms of Sko1 were not detectable after 1 min of treatment with 1,10-phenanthroline (Fig. 4A). Previous work showed that heat shock triggers elevated SUMO conjugation levels by increasing sumoylation and reducing desumoylation through partial depletion of Ulp1 (Moallem et al, 2023). We investigated whether heat-shock-triggered reduction in Ulp1 levels could offset the loss of SUMO conjugation after treatment with 1,10-phenanthroline. However, a 5-min treatment with 1,10-phenanthroline was still highly effective in eliminating SUMO conjugation, both overall and for Sko1 specifically, in heat-shock-treated cultures (Fig. 4B). Therefore, the increase in sumoylation that occurs during heat shock is not sufficient to reverse the effects of 1,10-phenanthroline.

Similar to Sko1, using a strain with HA-tagged Tfg1, we found that sumoylated forms of Tfg1 are virtually undetectable after 1 min of treatment with 1,10-phenanthroline (Fig. 4C). Tfg1 was also tagged with HA in the *ulp1-mt* strain and its WT parent (*ULP1*), allowing us to examine the effects of reduced Ulp1 activity on Tfg1 sumoylation dynamics. In the WT/*ULP1* background, heat shock resulted in increased Tfg1 sumoylation levels, but again, this did not impact the ability of 1,10-phenanthroline to virtually eliminate Tfg1 sumoylation (Fig. 4D). However, reduced Ulp1 activity in the *ulp1-mt* background completely reversed the effects of a 5-min 1,10-phenanthroline

treatment on Tfg1 sumoylation, in line with the idea that Ulp1 acts continuously to rapidly reverse SUMO conjugations. Like heat shock, ethanol stress triggers increased sumoylation levels, in this case by sequestering Ulp1 in the nucleolus (Sydorskyy et al, 2010). We examined whether nucleolar sequestration of Ulp1 during ethanol stress could reverse the rapid disappearance of SUMO conjugation after treatment with 1,10-phenanthroline. As seen in the whole cell lysate (Input) analysis in Fig. 4E, ethanol stress partially reversed the ability of the drug to eliminate SUMO conjugation. Ethanol treatment also modestly impacted the ability of 1,10-phenanthroline to reduce Tfg1 sumoylation, specifically (Fig. 4E; HA IP). Although ethanol treatment had only a partial effect, these results further implicate desumoylation by Ulp1 in the rapid removal of SUMO conjugates when the sumoylation machinery is impaired by 1,10-phenanthroline.

To provide additional support for this notion, we examined whether depletion of Ulp1 could reverse the effects of 1,10-phenanthroline. We generated a yeast strain that expresses an auxin-inducible degron fused to Ulp1 (Nishimura and Kanemaki, 2014). As shown in Fig. 4F, addition of the degron tag itself to the C-terminus of Ulp1 resulted in reduced effectiveness of 1,10-phenanthroline, and SUMO conjugations were further resistant to the drug when Ulp1 was depleted by the addition of auxin. These results strongly support the idea that Ulp1 is responsible for the rapid reversal of SUMO modifications, which only becomes apparent when de novo sumoylation is inhibited. In addition, these data demonstrate that the drug rapidly triggers the loss of sumoylation of two specific SUMO substrates.

Our findings show that 1,10-phenanthroline impedes the formation of the E1~SUMO intermediate and results in rapid inhibition of sumoylation. Its unprecedented ability to almost immediately eliminate the bulk of SUMO modifications suggests that it can be used widely to study the effects of sumoylation on other cellular processes, the interdependence of sumoylation and other PTMs, and the stability and dynamics of SUMO conjugation itself, overall and of individual target proteins. As ubiquitin conjugation is unaffected by treatment with 1,10-phenanthroline, the drug will be useful for distinguishing biological roles for the two analogous PTMs, and as we have shown for Sko1, Tfg1, and RanGAP1, 1,10-phenanthroline can be used to confirm that modified forms of proteins detected by immunoblot are derived from SUMO conjugation specifically. As with most small molecule drugs (Rao et al, 2019), 1,10-phenanthroline has secondary target effects, but we note that transcriptional inhibition by the drug is far less rapid and potent than its action against sumoylation. Nonetheless, parallel analysis with thiolutin treatment can serve as an

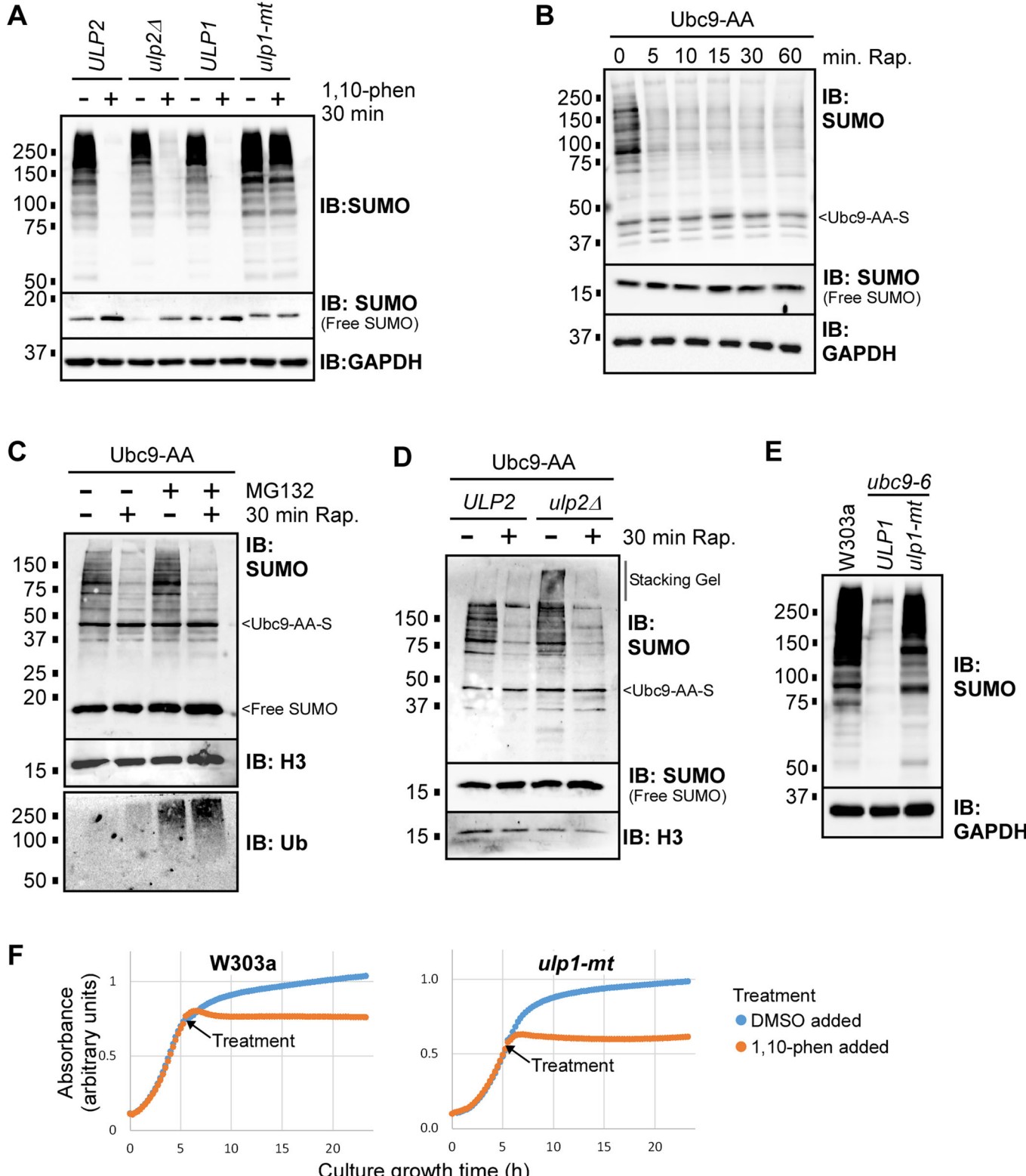

effective control for confirming that any effects of 1,10-phenanthroline are specific to inhibition of sumoylation, since thiolutin also impairs transcription but has no effect on SUMO conjugation levels. Sumoylation has recently been identified as a promising target for cancer immunotherapy because inhibiting sumoylation potentiates antitumour immunity (Chen, 2023; Gabellier et al, 2023). As such, it may be worthwhile to further explore the mechanisms and efficacy of SUMO inhibition in human cells by

**Figure 3. SUMO modifications are highly transient due to constitutive desumoylation by Ulp1.**

(A) Cultures of yeast strains lacking Ulp2 (*ulp2Δ*), expressing a partially inactive mutant form of Ulp1 (*ulp1-mt*), or their isogenic wild-type parental strains, *ULP2* and *ULP2*, respectively, were treated with 500 µg/mL 1,10-phenananthroline (+) or its solvent DMSO (−) for 30 min, then lysates were prepared and analyzed by SUMO and GAPDH immunoblots. A blot showing unconjugated SUMO peptides ("Free SUMO") is included. Defective processing of the SUMO precursor in the *ulp1-mt* strain leads to the detection of slower migrating free SUMO in samples from that strain (Li and Hochstrasser, 1999). (B) Cultures of the Ubc9-AA strain were treated with rapamycin for the indicated durations, then lysates were prepared and analyzed by SUMO and GAPDH immunoblots. Auto-sumoylated Ubc9-AA can be detected in the SUMO immunoblot ("Ubc9-AA-S"; Moallem et al, 2023). (C) The W303a strain was grown in conditions that permit inhibition of the 26 S proteasome by the drug MG132 (Liu et al, 2007). Cultures were then treated with MG132 (+) or its solvent DMSO (−), then with rapamycin (+) or its solvent DMSO (−), for 30 min before lysates were prepared and analyzed by SUMO, H3, and ubiquitin (Ub) immunoblots. H3 refers to histone H3 and serves as a loading control. (D) A strain was generated in which the *ULP2* gene was deleted in the Ubc9-AA background. Ubc9-AA strains expressing *ULP2* or with the *ulp2* knock-out were grown and treated with rapamycin or DMSO for 30 min as indicated. Lysates were then prepared and analyzed by SUMO and H3 immunoblots. (E) Cultures of the W303a strain, and strains expressing the *ubc9-6* mutant form of Ubc9 with wild-type (*ULP1*) or *ulp1-mt* forms of Ulp1 were grown, then lysates were prepared and analyzed by SUMO and GAPDH immunoblots. (F) Cultures of the W303a or *ulp1-mt* strains were grown starting from lag phase, and optical densities were determined at regular intervals as cultures grew to saturation. 1,10-phenanthroline (500 µg/mL) or DMSO were added at the treatment point indicated during the exponential phase. Average optical densities (absorbance) indicated are arbitrary units (see "Methods"). Source data are available online for this figure.

1,10-phenanthroline and related compounds. Additionally, in light of our findings, prior studies that used 1,10-phenanthroline as a sole method of inhibiting transcription should be re-evaluated to ensure that any effects that were observed are not attributable instead to the rapid elimination of sumoylation.

Intriguingly, using 1,10-phenanthroline and nuclear depletion of Ubc9, we found that SUMO conjugation is extremely transient in yeast, with most targets appearing to undergo nearly immediate desumoylation by Ulp1 once they are sumoylated. Our work directly supports a model that was suggested nearly 15 years ago in which sumoylation was proposed to occur in rapid "on-off" loops rather than functioning as a stable modification (Andreou and Tavernarakis, 2009). Why would cells continuously carry out this energy-consuming on-off cycle? As SUMO modifications preferentially occur on disordered regions of target proteins (Hendriks et al, 2017), perhaps the cycling itself, through yet unknown mechanisms, helps to stabilize these disordered regions,

or functions to regulate the availability of these regions for subsequent protein-protein interactions or for other types of PTMs. Alternatively, perhaps on-off sumoylation cycling primes cells for the rapid elevation of SUMO conjugation when needed, as observed during heat shock or ethanol exposure, by instantly destabilizing or sequestering SUMO proteases (Pinto et al, 2012; Lewicki et al, 2015; Moallem et al, 2023). Future studies can aim to explore the physiological benefits of rapid increases in sumoylation during the stress response and to uncover the purpose of the extreme transiency of this intriguing post-translational modification.

## Methods

### Reagents and Tools

See Table 1.

**Table 1. Reagents and tools.**

| Reagent/resource | Reference or source | Identifier or catalog number |
|---|---|---|
| *Antibodies* | | |
| Rabbit anti-yeast SUMO (Smt3; y-84) | Santa Cruz Biotechnology | sc-28649 |
| Rabbit anti-GAPDH | Sigma-Aldrich | G9545 |
| Rabbit anti-HA epitope | Sigma-Aldrich | H6908 |
| Mouse anti-HA agarose | Millipore | A2095 |
| Rat anti-RNA polymerase II subunit B1 (phospho CTD Ser-2) clone 3E10 | Sigma-Aldrich | 04-1571 |
| Rabbit anti-Histone H3 | Abcam | ab1791 |
| Mouse anti-multiubiquitin chain (Ub) clone FK1 | Cayman Chemical | 14219 |
| Mouse anti-SUMO1 | Developmental Studies Hybridoma Bank | SUMO1 21C7-s |
| Rabbit anti-SUMO2/3 | Abcam | ab22654 |
| Rabbit anti-Tetrahymena SUMO | James Forney (Purdue University) | Nasir et al, 2015 |
| IgG (H + L) Goat anti-rabbit HRP | Fisher Scientific | PI31460 |
| IgG (H + L) Goat anti-mouse HRP | Fisher Scientific | PI31430 |
| IgG (H + L) Goat anti-Rat HRP | Fisher Scientific | PI31470 |
| Rabbit anti-GAPDH | Sigma-Aldrich | G9545 |
| Rabbit anti-HA epitope | Sigma-Aldrich | H6908 |
| Mouse anti-HA agarose | Millipore | A2095 |
| Rat anti-RNA polymerase II subunit B1 (phospho CTD Ser-2) clone 3E10 | Sigma-Aldrich | 04-1571 |
| Rabbit anti-Histone H3 | Abcam | ab1791 |
| Mouse anti-multiubiquitin chain (Ub) clone FK1 | Cayman Chemical | 14219 |

**Table 1.** (continued)

| Reagent/resource | Reference or source | Identifier or catalog number |
|---|---|---|
| Mouse anti-beta actin | GenScript | RRID:AB_914102 |
| *Chemicals, enzymes, and other reagents* | | |
| Thiolutin | Enzo Life Sciences | BML-CT107 |
| 1,10-phenanthroline | Sigma-Aldrich | 131377 |
| 1,7-phenanthroline | Sigma-Aldrich | 301841 |
| Rapamycin | BioShop Canada | RAP004 |
| MG132 | Enzo Life Sciences | BML-PI102 |
| 3-indoleacetic acid (auxin) | Sigma-Aldrich | 45533 |
| Drop-out mix without Leu | US Biological | D9525 |
| Drop-out mix complete | US Biological | D9515 |
| Ulp1 protein (recombinant; yeast) | Abbkine | PRP3001 |
| Yeast nitrogen base without MgSO$_4$ | Sunrise Science | 1521-100 |
| TPEN (*N,N,N′,N′*-tetrakis(2-pyridinylmethyl)-1,2-ethanediamine) | Sigma-Aldrich | P4413 |
| *Critical commercial assays* | | |
| SUMOylation Assay Kit | Abcam | ab139470 |
| iScript cDNA Synthesis Kit | Bio-Rad | 1708891 |
| *Experimental models: cell lines* | | |
| Human: HeLa | ATCC | RRID:CVCL_0030 |
| Human: 293T | ATCC | RRID:CVCL_0063 |
| *Experimental models: organisms/strains* | | |
| *S. cerevisiae*: W303a [*MAT a ura3-52 trp1Δ2 leu2-3_112 his3-11 ade2-1 can1-100*] | Dharmacon Reagents | YSC1058 |
| *S. cerevisiae*: rpb1-1 [*MAT a ura3-52 rpb1-1 trp1::hisG*] | McNeil et al, 1998 | N/A |
| *S. cerevisiae*: YPH499 ("*No tag*" parent for Ubc9-HA and Uba2-HA strains) [*MATa ura3-52 lys2-801_amber ade2-101_ochre trp1-Δ63 his3-Δ200 leu2-Δ1*] | Sikorski and Hieter, 1989 | |
| *S. cerevisiae*: Ubc9-HA [*MATa UBC9-6HA::Kl TRP1 ura3-52 lys2-801_amber ade2-101_ochre trp1-Δ63 his3-Δ200 leu2-Δ1*] | This paper | |
| *S. cerevisiae*: Uba2-HA [*MATa UBA2-6HA::Kl TRP1 ura3-52 lys2-801_amber ade2-101_ochre trp1-Δ63 his3-Δ200 leu2-Δ1*] | This paper | |
| *S. cerevisiae*: BY4741 (*ULP2* parent) [*MATa his3Δ1 leu2Δ0 met15Δ0 ura3Δ0*] | Dharmacon Reagents | YSC1048 |
| *S. cerevisiae*: ulp2Δ [*MATa ulp2Δ::URA3MX his3Δ1 leu2Δ0 met15Δ0 ura3Δ0*] | This paper | N/A |
| *S. cerevisiae*: MH1006 (*ULP1* parent) [*MATa leu2-3,112 ura3-52 trp1-289 ade2Δ ade3Δ lys1::kanMX4*] | Soustelle et al, 2004 | N/A |
| *S. cerevisiae*: ulp1-mt [*MATa ulp1-I615N leu2-3,112 ura3-52 trp1-289 ade2Δ ade3Δ lys1::kanMX4*] | Soustelle et al, 2004 | N/A |
| *S. cerevisiae*: (Anchor-away parent strain) [*MATa tor1-1 fpr1::loxP-LEU2-loxP RPL13A-2×FKBP12::loxP*] | Euroscarf | HHY168 |
| *S. cerevisiae*: Ubc9-AA [*MATa UBC9-FRB::kanMX tor1-1 fpr1::loxP-LEU2-loxP RPL13A-2×FKBP12::loxP*] | Moallem et al, 2023 | N/A |
| *S. cerevisiae*: Ubc9-AA ulp2Δ [*MATa ulp2Δ UBC9-FRB::kanMX tor1-1 fpr1::loxP-LEU2-loxP RPL13A-2×FKBP12::loxP*] | This paper | N/A |
| *S. cerevisiae*: ubc9-6 [*MATa ubc9-1:Tadh1:TRP1:ubc9-1 ura3-52 trp1Δ2 leu2-3_112 his3-11 ade2-1 can1-100*] | Baig et al, 2021 | N/A |
| *S. cerevisiae*: ubc9-6 ulp1-mt [*MATa ulp1-I615N:URA3 ubc9-1:Tadh1:TRP1:ubc9-1 ura3-52 trp1Δ2 leu2-3_112 his3-11 ade2-1 can1-100]ubc9-1:Tadh1:TRP1:ubc9-1*] | This paper | N/A |
| *S. cerevisiae*: ubc9-6 ULP1 [*MATa ULP1:URA3 ubc9-1:Tadh1:TRP1:ubc9-1 ura3-52 trp1Δ2 leu2-3_112 his3-11 ade2-1 can1-100]ubc9-1:Tadh1:TRP1:ubc9-1*] | This paper | N/A |
| *S. cerevisiae*: Sko1-HA [*MATa SKO1-6HA::kl TRP1 ura3-52 trp1Δ2 leu2-3_112 his3-11 ade2-1 can1-100*] | Sri Theivakadadcham et al, 2019 | N/A |
| *S. cerevisiae*: Tfg1-HA [*MATa TFG1-6HA::kl TRP1 ura3-52 trp1Δ2 leu2-3_112 his3-11 ade2-1 can1-100*] | Baig et al, 2021 | N/A |
| *S. cerevisiae*: Tfg1-HA ULP1 [*MATa TFG1-6HA::kl TRP1 leu2-3,112 ura3-52 trp1-289 ade2Δ ade3Δ lys1::kanMX4*] | Baig et al, 2021 | N/A |
| *S. cerevisiae*: Tfg1-HA ulp1-mt [*MATa TFG1-6HA::kl TRP1 ulp1-I615N leu2-3,112 ura3-52 trp1-289 ade2Δ ade3Δ lys1::kanMX4*] | Baig et al, 2021 | N/A |

**Table 1.** (continued)

| Reagent/resource | Reference or source | Identifier or catalog number |
|---|---|---|
| *S. cerevisiae*: YMK728 (Parent AID strain) [*MAT a ura3-1::ADH1-OsTIR1(pMK200, URA3) trp1Δ2 leu2-3_112 his3-11 ade2-1 can1-100*] | NRBP Japan | BY26972 |
| *S. cerevisiae*: Ulp1-AID-HA [*MAT a ULP1-AID\*-6HA::Hyg ura3-1::ADH1-OsTIR1(pMK200, URA3) trp1Δ2 leu2-3_112 his3-11 ade2-1 can1-100*] | This paper | N/A |
| *T. thermophila*: CU428.2 (mating type VII) | Tetrahymena Stock Center, Cornell University | TSC_SD00178 |
| *Oligonucleotides* | | |
| *RPL25* RT-PCR forward ACCAACAGCCATTAACAAATCC | This paper | |
| *RPL25* RT-PCR reverse TTAACGACAGCTTTCTTAGCG | This paper | |
| *RPL27b* RT-PCR forward CTGGGTAATGATTTATCCTTC | This paper | |
| *RPL27b* RT-PCR reverse TGGGTGCAACAAATCTTGAAC | This paper | |
| 25 S RT-PCR forward TCTAGCATTCAAGGTCCCATTC | This paper | |
| 25 S RT-PCR reverse CCCTTAGGACATCTGCGTTATC | This paper | |
| *Recombinant DNA* | | |
| pHA-UB plasmid (pADH1-HA-Ub) | This paper | |
| LSP938-Gal1-HA-Ub | Gift from Linda Hicke; Addgene | 32175 |
| pGAD424 | Clontech | |
| pHyg-AID\*-6HA | Gift from Helle Ulrich; Addgene | 99520 |

## Methods and protocols

### Yeast strains and growth

*Saccharomyces cerevisiae* budding yeast strains used in this study are listed in the reagents and tools table (Table 1). Strains were grown in 10–50 mL of YPD medium (10 g/L yeast extract, 20 g/L peptone, 2% glucose) except when plasmid maintenance was required (i.e., Fig. 1B), in which case selective SC-Leu medium was used (1.74 g/L yeast nitrogen base without ammonium sulfate or amino acids, 5 g/L ammonium sulfate, 1.62 g/L drop-out mix without Leu, 2% glucose). Cultures were grown at 30 °C, unless otherwise noted, in orbital shaker incubators, oscillating at 200 rpm, to mid-log phase, generally at an absorbance of 0.5–1.0 at 595 nm, then treated or collected for further processing. Biological replicates were performed for all experiments.

### Construction of previously unreported yeast strains

For the detection of ubiquitinated substrates, the W303a strain was transformed with the pHA-Ub plasmid, which was constructed as follows. The HA-ubiquitin coding sequence was PCR amplified from plasmid LSP938-Gal1-HA-Ub (Addgene) and subcloned into vector pGAD424 (Clontech), immediately downstream of the *ADH1* promoter (replacing the Gal4 activation domain coding sequence). The Ubc9-HA and Uba2-HA strains were generated by insertion of the 6xHA::*Kl TRP1* PCR amplicon cassette at the *UBC9* or *UBA2* genomic loci in the YPH499 background strain through homologous recombination, as previously described (Knop et al, 1999). The *ulp2Δ* and *ulp2Δ* Ubc9-AA strains were generated through gene replacement by transforming host/parental strains

BY4741 and Ubc9-AA, respectively, with a URA3MX PCR amplicon containing 45-bp flanks that are homologous to sequences flanking the *ULP2* gene (Goldstein et al, 1999). The *ubc9-6 ULP1* and *ubc9-6 ulp1-mt* strains were generated by yeast transformation into the *ubc9-6* strain using PCR products that included the 3'-end of the *ULP1* coding region with the wild-type sequence or the I615N mutation, respectively, linked to the *URA3* reporter gene. Strains were confirmed by PCR, DNA sequencing, and/or immunoblot. The Ulp1-AID-HA strain was generated by insertion of the AID\*-6HA::Hyg PCR amplicon cassette at the *ULP1* genomic locus in the AID parental strain (YMK728) through homologous recombination, as previously described (Morawska and Ulrich, 2013).

### Treatment of yeast cultures

For treatment at 37 °C (i.e., heat shock), cultures were grown at 30 °C as indicated above, then moved to a 37 °C shaker for the indicated durations, while control cultures were left in the 30 °C shaker. For drug treatments, the following concentrations were used, unless otherwise noted: thiolutin, 4 µg/mL; 1,10-phenanthroline, 500 µg/mL; 1,7-phenanthroline, 500 µg/mL; rapamycin, 1 µg/mL; auxin (indole-3-acetic acid), 0.5 mM. Control cultures were treated with a volume of solvent (e.g., DMSO, ethanol) equal to the volume of drug added. For ethanol stress, ethanol was added to a final volume of 10%, while an equal volume of sterile water was added to control cultures. For treatment with MG132, overnight cultures were inoculated in SC +Pro medium (1.75 g/L yeast nitrogen base without ammonium sulfate or amino acids; 10 g/L proline; 2 g/L drop-out mix complete; 2% glucose). On the next morning, cultures were diluted

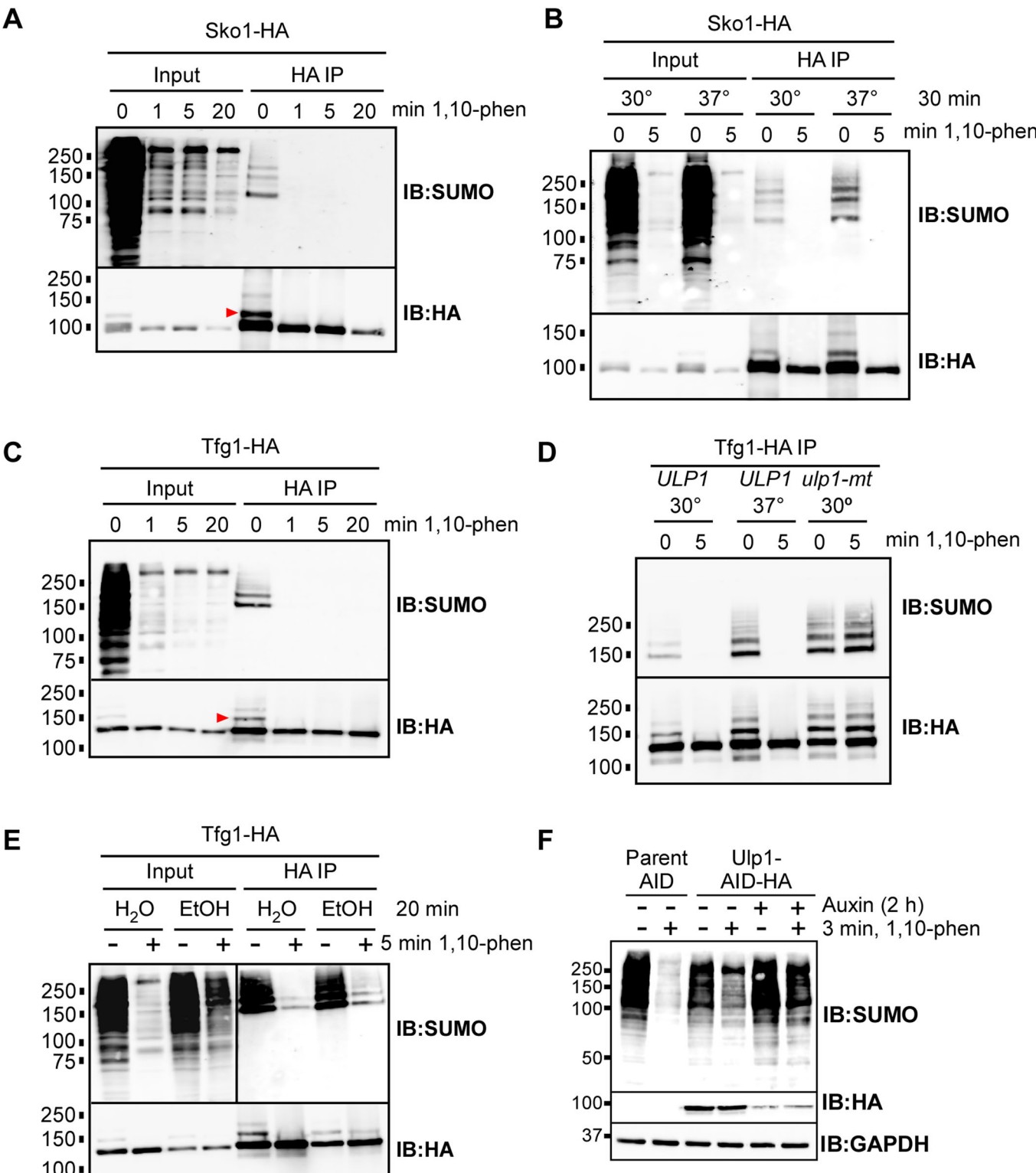

to a density of 0.5 (595 nm) in the same medium plus 0.003% SDS and grown for 3 h. Then, MG132 (to a final concentration of 75 µM) or an equal volume of DMSO was added, and cultures were grown for 30 min prior to further treatment (e.g., with rapamycin) and collection.

**Yeast liquid growth assays**

Exponentially growing cultures were diluted to an absorbance of ~0.2 at 595 nm, then transferred in triplicate to wells in a 96-well plate. The plate was inserted into an accuSkan absorbance microplate reader (Fisherbrand), which shook the samples regularly

**Figure 4.   Sumoylation of Sko1 and Tfg1 is highly transient but not stabilized by heat shock.**

(A) Cultures of a strain expressing HA-tagged Sko1 (Sko1-HA) were grown and treated with 500 μg/mL 1,10-phenanthroline for the indicated durations, then lysates were prepared, and an HA IP was performed. HA IPs and inputs (representing 5–10% of the material used for IP) were analyzed by SUMO and HA immunoblots. The red arrowhead in the HA immunoblot indicates the mono-sumoylated form of Sko1-HA, as previously described (Sri Theivakadadcham et al, 2019). (B) Cultures of the Sko1-HA strain were grown at the normal temperature of 30 °C, then either heat shocked by transferring to 37 °C for 30 min or left at 30 °C, then treated with 500 μg/mL of 1,10-phenanthroline or DMSO ("0 min") for 5 min. HA IPs were performed then analyzed by SUMO and HA immunoblots. (C) Cultures of a strain expressing an HA-tagged version of Tfg1, the large subunit of general transcription factor TFIIF, were grown and treated with 500 μg/mL 1,10-phenanthroline for the indicated durations. Lysates were then prepared, and HA IPs were performed. HA IPs and inputs were then analyzed by SUMO and HA immunoblots. The red arrowhead in the HA immunoblot indicates mono-sumoylated Tfg1, as previously described (Baig et al, 2021). (D) A strain expressing Tfg1-HA in the *ulp1-mt* background strain, or its isogenic parent strain (*ULP1*) were generated. Cultures of the strains were grown at the normal temperature of 30 °C, then either heat shocked at 37 °C for 30 min or left at 30 °C, then treated with 500 μg/mL of 1,10-phenanthroline or DMSO ("0 min") for 5 min. HA IPs were performed then analyzed by SUMO and HA immunoblots. (E) Cultures of the Tfg1-HA stain were treated with 10% ethanol or the same volume of H$_2$O for 20 min, then exposed to 1,10-phenanthroline or DMSO ("0 min") for 5 min. Lysates were then prepared and HA IPs were analyzed by SUMO and HA immunoblots. (F) A strain expressing HA-tagged Ulp1 fused to an auxin-induced degron (AID) and its parental strain were grown in culture and treated with auxin (indole-3-acetic acid) or an equal volume of ethanol for 2 h, then treated with 500 μg/mL of 1,10-phenanthroline, or control-treated with DMSO, for 3 min. Lysates were immediately prepared and analyzed by SUMO, HA and GAPDH immunoblot. Source data are available online for this figure.

and maintained a temperature of 30 °C while taking absorbance measurements (595 nm) at 15-min intervals over a period of 20 h. When cultures were at mid-log phase (shortly after 5 hours of growth), measurements were paused while 1,10-phenanthroline or DMSO were added, as appropriate, then readings were resumed. Wells containing un-inoculated medium were used for background measurements. Background-subtracted average absorbance readings and standard deviations are provided in the source data. Because the light path distance in the accuSkan is less than 1 cm, absorbance measurements are indicated as arbitrary units (a.u.) and are not directly comparable to values obtained through a standard spectrophotometer.

### Preparation of yeast lysates

When ready, cultures were briefly chilled on ice, then harvested by centrifugation at 3000 g at 4 °C for 5 min. Supernatants were discarded and pellets were washed with IP buffer (50 mM Tris-HCl, pH8; 150 mM NaCl; 0.1% Nonidet P-40 (NP40); 0.1 mM DTT (except where indicated); supplemented with a protease inhibitor cocktail for yeast (Sigma-Aldrich), 1 mM phenylmethylsulphonyl fluoride (PMSF), and 2.5 mg/mL N-ethylmaleimide (NEM)) that was pre-chilled on ice, using a volume equal to the original culture volume. Washed pellets were then resuspended in 500 μL of IP buffer and 0.25 g of acid-washed glass beads (425–600 μm; Sigma-Aldrich) were added. Samples were vortexed at 4 °C for 15 min, placed on ice for 5 min, vortexed again for 15 min, then lysates were transferred to fresh microfuge tubes by micropipette. Lysates were then clarified by centrifuging at 4 °C at maximum microfuge speed, transferring supernatants to fresh tubes, then repeating the centrifugation and transfer. Clarified lysates were then used for immunoblot or immunoprecipitation analysis.

### Immunoblot and immunoprecipitation

For protein samples to be analyzed by immunoblot, an equal volume of 2× sample buffer (140 mM Tris-HCl, pH 6.8; 4% SDS; 20% glycerol; 0.02% bromophenol blue; supplemented with 10% 2-mercaptoethanol prior to use, except when nonreducing conditions were needed) was added, then the mixtures were boiled for 4 min. Prior to electrophoresis, samples were centrifuged to eliminate debris, then supernatants were separated by SDS-PAGE using 7.5% or 10% resolving gels. For most immunoblots, separated proteins were then transferred to nitrocellulose membranes by tank transfer at 100 V for 1 hr in a buffer consisting of 50 mM Tris-HCl, 380 mM glycine, 0.1% SDS, and 10% methanol for detection of large proteins (e.g., SUMO conjugates), or

25 mM Tris-HCl, 190 mM glycine, and 20% methanol for detection of small proteins, including free SUMO. Blots were then blocked in 5% skim milk in PBST (1× PBS, 0.05% Tween) then probed with primary antibodies overnight, rocking at 4 °C. Blots were then washed in PBST, probed with secondary antibodies for 1 h at room temperature, then chemiluminescence-based imaging was performed with a MicroChemi imager (DNR). Antibodies for immunoblots, except where otherwise specified, were used as follows: 1:500–1:1000 yeast SUMO (Smt3; y-84; Santa Cruz); 1:3000 GAPDH (Sigma-Aldrich); 1:2000 HA (12CA5; Sigma-Aldrich); 1:2000 RNAPII Ser2p (3E10; Sigma-Aldrich); 1:1000 SUMO1 (Abcam); 1:1000 SUMO2/3 (Abcam); 1:3000 histone H3 (Abcam); 1:1000 Ub (FK1; Cayman Chemical).

For immunoprecipitations, 15 μL of anti-HA-agarose beads (Sigma-Aldrich) were added to clarified lysates, and samples were rotated overnight at 4 °C. Beads and bound material were washed twice with IP buffer, then twice in IP buffer lacking NP40. Beads were then resuspended in 100 μL of 2× sample buffer and boiled for 4 min to elute bound proteins. Samples were centrifuged to eliminate beads debris, and supernatants were then analyzed by immunoblot as indicated above.

### In vitro sumoylation analysis

In vitro sumoylation reactions were performed using the Sumoylation Assay Kit (Abcam), which includes human proteins SAE1/UBA2, UBC9, SUMO1, SUMO2, and as a sample substrate, RanGAP1. For analysis of UBC9 auto-sumoylation, RanGAP1 was excluded. Standard reactions were performed as suggested by the supplier and included 5 mM of Mg/ATP (i.e., 5 mM each of MgCl$_2$ and ATP), but this was adjusted or omitted where indicated. Where indicated, reactions were treated with DMSO, 1,10-phenanathroline, or 1,7-phenanthroline prior to incubation. Reactions were carried out for 1 h at 37 °C, then analyzed by SDS-PAGE and immunoblot analysis as described above.

### RT-qPCR analysis

For analysis of RNA levels, 10-mL yeast cultures were grown, and RNA was isolated and analyzed by RT-qPCR as previously described (Baig et al, 2021; Sri Theivakadadcham et al, 2019). qPCR was performed using primers indicated in the Reagents and Tools table in a CFX Opus real-time PCR detection system (Bio-Rad).

### Human cell culture, lysate preparation, and analysis

Human 293T and HeLa cell cultures were maintained in 6-cm dishes with DMEM supplemented with 10% fetal bovine serum, then treated with 2.5 mg/mL of 1,10-phenanthroline or with DMSO

in serum-free DMEM. Medium was discarded and cells were washed with ice-cold PBS. Cells were scraped and chilled lysis buffer (0.25 mL per plate; 150 mM NaCl, 1% Triton X-100, 0.5% sodium deoxycholate, 50 mM Tris-HCl, pH 8.0; 2.5 mg/mL *N*-ethylmaleimide) was added to the cells, which were then scraped and transferred to microfuge tubes. For the 30-min treated samples, cells became less adherent and were therefore transferred to microfuge tubes prior to washing in PBS and the addition of the lysis buffer. Samples were placed on a rotator at 4 °C for 30 min, then spun at maximum speed in a refrigerated microfuge (4 °C). Supernatants were transferred to fresh tubes and analyzed by SUMO1 and SUMO2/3 immunoblots. Antibodies were used at the following dilutions: 1:250 SUMO1 (DSHB); 1:125 SUMO2/3 (Abcam). For loading control, lysates separated by SDS-PAGE were stained with Coomassie brilliant blue.

### Tetrahymena culture, lysate preparation, and analysis

Wild-type *Tetrahymena thermophila* strain CU428.2 was grown at 30 °C, shaking at 100 rpm, in super proteose peptone (SPP) medium containing 1% proteose peptone (Becton Dickinson, Sparks, MD, USA), 0.1% yeast extract (Becton Dickinson), 0.2% glucose (Sigma-Aldrich, St. Louis, MO, USA), and 0.003% EDTA-Fe (Sigma-Aldrich). Cells were untreated or treated with 500 μg/mL or 2.5 mg/mL of 1,10-phenanthroline for 1, 5, or 30 min. Cells were collected, then fixed with 10% (w/v) trichloroacetic acid (TCA; Sigma-Aldrich) and incubated on ice for 30 min. After removal of TCA by centrifugation at 12,000× *g* for 2 min, cell pellets were lysed in PAGE sample buffer (2% SDS ([Sigma-Aldrich], 2.5% 2-mercaptoethanol [Sigma-Aldrich], 10% glycerol [Sigma-Aldrich], and 50 mM Tris-HCl, pH 6.8) and boiled for 5 min prior to analysis by SDS-PAGE and immunoblot. Antibodies used were anti-Tetrahymena SUMO (1:1000; Gift from Jim Forney), or anti-β-actin antibody (1:1000; GenScript, Piscataway, NJ, RRID:AB_914102).

## Data availability

No novel datasets were produced in this study.

## Peer review information

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

## Acknowledgements

We thank Benjamin Bergey, Farhan Khan, and Akhi Akhter for technical assistance, Ronald Pearlman for assistance in securing reagents, and James Forney (Purdue University) for generously providing the Tetrahymena SUMO antibody. Research in the Rosonina lab is funded by grants from the Canadian Institutes of Health Research (CIHR; grant numbers MOP-142282 and PJT-178112) and the Natural Sciences and Engineering Research Council of Canada (NSERC; RGPIN-04208-2014) to ER.

## Author contributions

**J Bryan McNeil:** Conceptualization; investigation; methodology; writing—original draft; writing—review and editing. **Su-Kyong Lee:** Investigation; methodology. **Anna Oliinyk:** Investigation; methodology. **Sehaj Raina:** Investigation; methodology. **Jyoti Garg:** Investigation; methodology. **Marjan Moallem:** Investigation; methodology. **Verne Urquhart-Cox:** Conceptualization; methodology. **Jeffrey Fillingham:** Resources; supervision; writing—review and editing. **Peter Cheung:** Resources; supervision; writing—review and editing. **Emanuel Rosonina:** Conceptualization; resources; formal analysis; supervision; funding acquisition; writing—original draft; writing—review and editing.

## Disclosure and competing interests statement

The authors declare no competing interests.

