## [Peer Review File · EMBO Reports]

1,10-phenanthroline inhibits sumoylation and reveals that yeast SUMO modifications are highly transient

J. Bryan McNeil, Su-Kyong Lee, Anna Oliinyk, Sehaj Raina, Jyoti Garg, Marjan Moallem, Verne Urquhart-Cox, Jeffrey Fillingham, Peter Cheung, and Emanuel Rosonina

DOI: [10.15252/embr.202357004](https://doi.org/10.15252/embr.202357004)

Corresponding author(s): Emanuel Rosonina (rosonina@yorku.ca)

Review Timeline:

Presubmission Date:	14th Feb 23
Presubmission Editorial Decision:	17th Feb 23
Revision Received:	27th Feb 23
Editorial Decision:	26th Apr 23
Appeal Received:	7th Sep 23
Editorial Decision:	14th Sep 23
Revision Received:	16th Sep 23
Editorial Decision:	27th Oct 23
Revision Received:	31st Oct 23
Accepted:	13th Nov 23

Transaction Report:

Dear Dr. Rosonina,

Thank you for enquiring about the potential suitability of your manuscript for EMBO Reports.

I have now read your abstract and cover letter and discussed it with the team and an expert advisor and we all consider your data on the short-lived SUMOylation in yeast and the novel inhibitor of Ubc9 interesting and potentially a good fit for EMBO reports.

I therefore invite you to submit the full manuscript to have it peer reviewed at EMBO reports.

I am looking forward to receiving your manuscript.

Yours sincerely,

To submit your manuscript, please follow this link.

Link Not Available

25th Apr 2023

Dear Dr. Rosonina,

Thank you for the submission of your manuscript to EMBO reports. I apologize for the delay in handling your manuscript, but we have meanwhile received the two enclosed reports on it.

I am sorry to say that we cannot offer publication in EMBO reports. While referee 1 is more positive, we also note the concerns from referee 2 regarding the advance provided and the broader applicability of 1,10-phenanthroline, which limit the broader significance of your study to the readers of our journal.

While we cannot pursue this manuscript further at EMBO reports, we encourage you to transfer your study to our not-for-profit open-access sister journal, Life Science Alliance (LSA).

We shared your manuscript and the accompanying reviews with LSA Executive Editor, Eric Sawey, who is interested in these findings, and would like to invite further consideration of this manuscript at LSA pending the following revisions:

- Address Reviewer 1's comments.
- Address Reviewer 2's Major points #2, 3 & 5, point #4 via added Discussion, and the Minor points.

We encourage you to use the link below to transfer your manuscript to LSA. You do not need to revise the manuscript before transferring it to LSA. Once you transfer, Dr. Sawey will email you an invitation to revise and resubmit, listing the same revision requests as mentioned above. Please feel free to reach out at e.sawey@life-science-alliance.org if you have any questions about the LSA journal, the transfer process or the revisions requested.

Kind regards,

Referee #1:

McNeil et al. demonstrate that treatment of yeast cells with 1,10-phenanthroline leads to a strikingly fast loss of SUMO conjugates. This effect is faster than those on transcription. They further show that a functional Ulp1 SUMO protease is required for the rapid disappearance of SUMO conjugates upon treatment with 1,10-phenanthroline. In vitro, they show that formation of a Ubc9-SUMO thioester intermediate is inhibited by 1,10-phenanthroline. Surprisingly, treatment of HeLa cells with 1,10-phenanthroline leads to a loss of SUMO1 conjugates, whereas no such effect is observed in HEK 293T cells.

This is a very well conducted and controlled study with interesting results, worthy of publication in EMBO Reports in my opinion. Even though 1,10-phenanthroline has a wider range of physiological effects, as acknowledged by the authors, the utilization of 1,10-phenanthroline as a rapid inhibitor of SUMO protein modification will be a helpful experimental approach in studying this PTM. The results highlight the rapid action of the SUMO protease Ulp1 under the applied conditions.

I have only a couple of small specific comments:

The notion that Ulp1 action would be "constant and constitutive" may be slightly misleading as Ulp1 is known to shuttle from a NPC localization (doi: 10.1007/s004380000254) to a cytosolic one (DOI: 10.1186/1741-7007-9-74) within different stages of the cell cycle, or translocates to the nucleolus upon ethanol treatment as mentioned by the authors. The relevance of these or other regulatory inputs acting upon Ulp1 in a physiological context seems neglected with the above-mentioned wording.

Based upon the data, it is a maybe remote, but formal possibility that the disappearance of SUMO conjugates upon treatment of yeast cells with 1,10-phenanthroline is due to increased Ulp1 activity rather than a decrease in conjugation. Consistent with the former possibility would be that there is very little difference in the SUMO pattern of the *ulp1*-mt mutant whether or not it has been treated with 1,10-phenanthroline for 30 min (Fig. 3A). To distinguish between the two possibilities would probably require a more complicated experimental setup with a detection of de novo sumoylation during treatment, e.g. in a pulse chase setting. This could be discussed.

Referee #2:

In this manuscript, McNeil et al. report an investigation of the effects of a known transcription inhibitor, 1,10-phenanthroline, on SUMO conjugation. They find that the compound rapidly inhibits SUMO conjugation in yeast cells and trace this effect to an early step of the SUMO conjugation cascade. Using this inhibitor, they then show that most SUMOylation in yeast is very transient, due to constitutive activity of the major SUMO-specific isopeptidase, Ulp1.

The experiments described here are largely clean and convincing (with some minor exceptions as detailed below). The findings confirm the long-standing notion in the field that SUMOylation generally undergoes rapid recycling, although details about how rapid the turnover is have not been reported to my knowledge. The mechanistic insight as well as the practical utility remain limited, as discussed below, and it appears as if the authors had trouble deciding whether to write it as a technical contribution ('1,10-phen. is a useful inhibitor of SUMOylation in yeast') or a mechanistic study ('SUMOylation is transient and curbed by Ulp1'). As it stands, the manuscript falls a bit short on both counts. As the authors explain themselves, the utility of the compound as a SUMOylation inhibitor is limited to very special circumstances where effects are measured within a few minutes (i.e. before other effects of the compound set in).

Major points:

1. The mechanism by which 1,10-phen. inhibits the SUMO system is not pursued in great detail. If the authors would like to frame this manuscript as a mechanistic study, it would be necessary to show whether the compound inhibits the E1 or the E2, and provide more detailed kinetic data in a homogeneous system, i.e. with all yeast and/or all human enzymes.
2. The effects on transcription should be more carefully compared to the effects on SUMO conjugation. The authors argue that inhibition of SUMOylation is much faster than transcription inhibition; yet, they show only limited data on transcription, reflected by a blot against the S2p form of RNAPII. I think it would be necessary to perform some assay with different times and doses for nascent transcription to support that statement.
3. The statement that toxicity of the inhibitor is not due to its effect on SUMOylation appears premature because it is also possible that an interference with rapid deSUMOylation is incompatible with long-term survival.
4. The effects of the compound on mammalian cells are poorly characterized. A detailed study may be beyond the scope of this manuscript, but the differences between the SUMO isoforms are interesting and call for an explanation. Are there differences in the human conjugation system that explain this (could be tested in vitro) or is the kinetics of SUMO1 versus SUMO2/3 conjugates different in cells, perhaps due to the activity of isoform-specific isopeptidases? This would be relevant and interesting questions to ask.
5. Given the role of Ulp1 as a processing enzyme for the SUMO precursor, is it possible that the synthetic lethal interaction between AA-Ubc9 and the Ulp1 mutant is due to an overall insufficient level of mature SUMO, i.e. would the expression of processed SUMO rescue the effect?

Minor points:

1. The treatment times should be more precisely noted in some of the figures, e.g. Figure 1 and S1.
2. RT-PCR controls should be performed not only for thiolutin, but also for 1,10-phen.
3. The switch between molar and mass-based concentrations for inhibitor concentration is annoying, as it makes comparisons difficult.
4. Fig. 3B: rapamycin treatment should also be performed on WT cells as a control.

** As a service to authors, EMBO Press provides authors with the ability to transfer a manuscript that one journal cannot offer to publish to another journal, without the author having to upload the manuscript data again. To transfer your manuscript to another EMBO Press journal using this service, please click on Link Not Available

Dear Dr. Rembold,

Thank you for having our manuscript reviewed and for suggesting *Life Science Alliance* as an alternate venue for its publication. However, as reiterated below, our paper presents two striking advances regarding sumoylation and its analysis, supported by solid data. As such, I feel strongly that the manuscript meets the aims and scopes of *EMBO Reports* and would benefit greatly from the journal's broad readership as a Scientific Report. We have generated new data that address all comments of both reviewers, and based on the information below, we request that you reconsider the manuscript for possible publication in *EMBO Reports*.

1. In your letter, you cited the concerns from Referee 2 regarding the advance provided and the broader applicability of our finding. I would like to reiterate the two major findings that we present, which we now emphasize more explicitly in the revised manuscript.

First, we demonstrate for the first time that SUMO modifications are extremely transient. As noted by Referee 2, some have speculated that SUMO modifications are transient, but to my knowledge, our work is the first to unambiguously demonstrate this and just how remarkably transient the modifications are. This surprising result alters the way that we view this post-translational modification, supporting an intriguing model in which SUMO modifications undergo constant conjugation-deconjugation cycles, and has fundamental implications for the study of SUMO biology.

Secondly, we report on a previously unknown, highly potent inhibitor of sumoylation (1,10-p) that works in vitro and in living cells. In contrast to the view of Referee 2, this will have wide applicability for the study of sumoylation and its dynamics/kinetics. Our new manuscript now highlights this more explicitly, and example applications are listed. We also note that inhibition of sumoylation is now considered a promising target for facilitating immuno-oncology therapy, and as such our finding will likely be of interest to those seeking methods for inhibiting sumoylation to potentiate cancer immunotherapy.

2. We have addressed all the comments of the reviewers, as briefly summarized here for your consideration at this stage. (A complete rebuttal will be available).

Reviewer 1: Only minor points were suggested

1. *Does 1,10-p activate Ulp1?*

In new in vitro data, we now demonstrate that 1,10-p does not enhance desumoylation by Ulp1.

Reviewer 2: Major Points

2.1 *It would be necessary to show whether the compound inhibits the E1 or E2.*

We now show through in vivo analysis that 1,10-p inhibits the E1 enzyme, specifically.

2.3 *Effects on transcription should be more carefully compared to effects on sumoylation.*

We now include a comparison of the rates of inhibition of transcription and sumoylation over a time course of treatment with 1,10-phenanthroline. The effects on sumoylation are notably more dramatic.

2.3 Statement that toxicity is not due to effects on sumoylation is premature.

We have altered the statement to address this point.

2.4 Effects on mammalian cells are poorly characterized (a detailed study may be beyond the scope of this manuscript).

We agree that a detailed study is beyond the scope of this manuscript. However, we have now better characterized the effects and show that 1,10-p is an effective inhibitor of sumoylation in two human cell lines. Our new data shows that 1,10-p impacts SUMO1 and SUMO2/3 conjugation in a similar manner in both cell lines, which clarifies our previous findings.

2.5 Is the synthetic lethal interaction between AA-Ubc9 and ulp1-mt due to insufficient free SUMO levels?

We now explain why we don't think this is the case.

Reviewer 2: Minor Points

All have been addressed.

In summary, we are pleased that Referee 1 found our study worthy of publication in *EMBO Reports* and we have addressed all concerns of the second reviewer. Our new data now bolster our findings, and, critically, the revised manuscript more explicitly emphasizes the key advances that our study provides.

Thank you for considering our request for reconsideration. If you are in agreement, we will submit a formal response letter and an updated manuscript.

Take care,
Emanuel

Emanuel Rosonina, PhD
Associate Professor
York University | Biology
Toronto, Canada
rosonina.lab.yorku.ca
416-736-2100 x44702

Dear Dr. Rosonina

Thank you for your message asking us to reconsider our decision and invite revision of your manuscript. I am sorry that it took me so long to come back to you with an answer but I have now read your letter, re-read the referee reports, and discussed the matter again with the editorial team.

I appreciate that you seem to have addressed all technical concerns from the referees. Given the work you have invested and given the support from at least one referee, we have no objection to consider the revised version for potential publication in EMBO Reports. I should however note that the concern regarding the broader utility of 1,10-phenanthroline to study the effect of SUMOylation remains an important point for potential publication here.

I am not including our instructions for submitting revised manuscripts here as these editorial points can be sorted out in case the referee reports are positive. But please adhere to the points I list below:

We need:

2) individual production quality figure files as .eps, .tif, .jpg (one file per figure). See https://wol-prod-cdn.literatumonline.com/pb-assets/embo-site/EMBOPress_Figure_Guidelines_061115-1561436025777.pdf for more info on how to prepare your figures).

3) Before submitting your revision, primary datasets produced in this study need to be deposited in an appropriate public database (see <https://www.embopress.org/page/journal/14693178/authorguide#datadeposition>). Please remember to provide a reviewer password if the datasets are not yet public. The accession numbers and database should be listed in a formal "Data Availability" section placed after Materials & Method (see also <https://www.embopress.org/page/journal/14693178/authorguide#datadeposition>). Please note that the Data Availability Section is restricted to new primary data that are part of this study.

* Note - All links should resolve to a page where the data can be accessed. *

I am looking forward to receiving your revised manuscript.

Yours sincerely,

Response to Reviewers

We thank both reviewers for their insightful comments and suggestions and for their careful analysis of our work. Our responses to the reviewers' comments are in bold.

Referee #1

McNeil et al. demonstrate that treatment of yeast cells with 1,10-phenanthroline leads to a strikingly fast loss of SUMO conjugates. This effect is faster than those on transcription. They further show that a functional Ulp1 SUMO protease is required for the rapid disappearance of SUMO conjugates upon treatment with 1,10-phenanthroline. In vitro, they show that formation of a Ubc9-SUMO thioester intermediate is inhibited by 1,10-phenanthroline. Surprisingly, treatment of HeLa cells with 1,10-phenanthroline leads to a loss of SUMO1 conjugates, whereas no such effect is observed in HEK 293T cells.

This is a very well conducted and controlled study with interesting results, worthy of publication in *EMBO Reports* in my opinion. Even though 1,10-phenanthroline has a wider range of physiological effects, as acknowledged by the authors, the utilization of 1,10-phenanthroline as a rapid inhibitor of SUMO protein modification will be a helpful experimental approach in studying this PTM. The results highlight the rapid action of the SUMO protease Ulp1 under the applied conditions.

We appreciate that the reviewer finds the study interesting, worthy of publication in *EMBO Reports*, and recognizes the usefulness of 1,10-p as a rapid inhibitor of sumoylation in studying the modification.

I have only a couple of small specific comments:

The notion that Ulp1 action would be "constant and constitutive" may be slightly misleading as Ulp1 is known to shuttle from a NPC localization (doi: 10.1007/s004380000254) to a cytosolic one (DOI: 10.1186/1741-7007-9-74) within different stages of the cell cycle, or translocates to the nucleolus upon ethanol treatment as mentioned by the authors. The relevance of these or other regulatory inputs acting upon Ulp1 in a physiological context seems neglected with the above-mentioned wording.

This is a good point, and we have now changed the terminology to "continuous" and specified that it acts this way "at least under normal growth conditions" (p. 9). I think this better expresses how, under normal growth conditions, Ulp1 acts as part of the continuous sumoylation-desumoylation cycle, but under certain conditions, Ulp1 activity can be reduced by different strategies.

Based upon the data, it is a maybe remote, but formal possibility that the disappearance of SUMO conjugates upon treatment of yeast cells with 1,10-phenanthroline is due to increased Ulp1 activity rather than a decrease in conjugation. Consistent with the former possibility would be that there is very little difference in the SUMO pattern of the *ulp1-mt* mutant whether or not it has been treated with 1,10-phenanthroline for 30 min (Fig. 3A). To distinguish between the two possibilities would probably require a more complicated experimental setup with a detection of de novo sumoylation during treatment, e.g. in a pulse chase setting. This could be discussed.

To address this possibility, we determined whether 1,10-p can increase Ulp1 activity by treating yeast lysates with dilute concentrations of recombinant Ulp1 and observing whether addition of 1,10-p enhances Ulp1's desumoylation activity. As we show in new Fig. S1F, 1,10-p does not increase Ulp1 activity (in fact it somewhat decreases the activity at higher 1,10-p concentrations). Additionally, we include new data that show that 1,10-p inhibits formation of the E1~SUMO thioester intermediate in

Response to Reviewers

yeast (new Fig. 2E). Together with our in vitro sumoylation analysis in which a desumoylase is not present, all together, the data point to a mechanism in which treatment with 1,10-p inhibits the first step of sumoylation thereby preventing de novo sumoylation events, and the rapid disappearance of SUMO conjugates is due to the activity of Ulp1. This is also supported by new data that show that, like in the *ulp1-mt* strain, depletion of Ulp1 (by an auxin-induced degron fusion) also reduces the effect of 1,10-p on reducing existing SUMO levels in yeast (new Fig. 4F).

Referee #2:

In this manuscript, McNeil et al. report an investigation of the effects of a known transcription inhibitor, 1,10-phenanthroline, on SUMO conjugation. They find that the compound rapidly inhibits SUMO conjugation in yeast cells and trace this effect to an early step of the SUMO conjugation cascade. Using this inhibitor, they then show that most SUMOylation in yeast is very transient, due to constitutive activity of the major SUMO-specific isopeptidase, Ulp1.

The experiments described here are largely clean and convincing (with some minor exceptions as detailed below). The findings confirm the long-standing notion in the field that SUMOylation generally undergoes rapid recycling, although details about how rapid the turnover is have not been reported to my knowledge. The mechanistic insight as well as the practical utility remain limited, as discussed below, and it appears as if the authors had trouble deciding whether to write it as a technical contribution ('1,10-phen. is a useful inhibitor of SUMOylation in yeast') or a mechanistic study ('SUMOylation is transient and curbed by Ulp1'). As it stands, the manuscript falls a bit short on both counts.

We appreciate that the reviewer recognizes the novelty of our finding that SUMO modifications are exceptionally transient. We feel that we have made two striking novel observations in this study: (1) identification of a sumoylation inhibitor that nearly abolishes modifications within a minute, and (2) most SUMO modifications appear to last only seconds. Identifying the inhibitor was critical for determining that sumoylation is so transient, and consequently the two stories are linked and presented together. We think that these findings are significant and will be of interest to many people in the field, and feel that reporting them, supported with clean and convincing data (including new experiments that address referee comments), as a short article in *EMBO Reports* will be beneficial at this stage. We anticipate that our work will lead to further studies exploring in more detail the mechanism of inhibition by 1,10-p and the significance of the transiency of sumoylation.

As the authors explain themselves, the utility of the compound as a SUMOylation inhibitor is limited to very special circumstances where effects are measured within a few minutes (i.e. before other effects of the compound set in).

We have expanded our discussion of the usefulness of 1,10-p for studying aspects of sumoylation in the Discussion (p. 13). Among the applications that we now describe, we think the drug can be used widely to study the effects of sumoylation on other processes and on its interaction with other PTMs, for examining the stability and dynamics of SUMO conjugation, and for distinguishing between ubiquitin vs. SUMO modifications and their effects. Most small molecule inhibitors have secondary or off-target effects, but that does not necessarily preclude them from being useful. We anticipate that this will also be the case for 1,10-p. Nonetheless, although 1,10-p works far more potently and rapidly as an inhibitor of sumoylation than of transcription (as we now show in new Fig. 1E), we suggest the

Response to Reviewers

use of thiolutin treatment in parallel to identify effects of 1,10-p that are specific to sumoylation, since thiolutin also inhibits transcription but does not affect SUMO conjugation levels. Additionally, we note that inhibition of sumoylation is recently considered a promising strategy for potentiating antitumour immunity, and therefore our findings may be of value for those seeking to identify novel mechanisms of inhibiting the modification.

Major points:

1. The mechanism by which 1,10-phen. inhibits the SUMO system is not pursued in great detail. If the authors would like to frame this manuscript as a mechanistic study, it would be necessary to show whether the compound inhibits the E1 or the E2, and provide more detailed kinetic data in a homogeneous system, i.e. with all yeast and/or all human enzymes.

To address this point, we examined whether formation of the thioester-linked E1~SUMO intermediate is affected by treatment of yeast cultures with 1,10-p. As shown in new Fig. 2E, formation of the E1~SUMO species (Uba2~SUMO, specifically) is dramatically reduced upon addition of 1,10-p. This supports the notion that it is E1 activity specifically that is inhibited by 1,10-p. As such, 1,10-p strongly inhibits the first step of sumoylation, thereby effectively preventing de novo sumoylation from taking place.

Towards further uncovering the mechanism of inhibition by 1,10-p, we now provide additional support that magnesium chelation is not involved. Specifically, growth of cultures in medium containing 10x less magnesium than normal does not enhance 1,10p activity (Fig. S1B). Furthermore, we now show that treatment of cultures with a commonly used zinc chelating agent, TPEN, does not impact SUMO conjugation levels (new Fig. S1C), arguing that zinc depletion is not the mechanism by which 1,10-p reduces sumoylation. Finally, we are eager to understand more about the molecular mechanisms of inhibition by 1,10-p, but feel that a detailed kinetic analysis is beyond the scope of this study.

2. The effects on transcription should be more carefully compared to the effects on SUMO conjugation. The authors argue that inhibition of SUMOylation is much faster than transcription inhibition; yet, they show only limited data on transcription, reflected by a blot against the S2p form of RNAPII. I think it would be necessary to perform some assay with different times and doses for nascent transcription to support that statement.

We performed the experiment suggested by the reviewer. RT-qPCR analysis was performed, using intron-exon spanning primers for two ribosomal protein genes, whose splicing is very rapid, and for which levels of intron-containing transcripts are therefore a reasonable measure of nascent transcription. As shown in new Fig. 1 E, a time-course of 1,10-p treatment was performed. Inhibition of nascent transcription by the drug is significantly slower and less effective than its immediate impact on sumoylation. Additionally, for comparison, we indicate the range of concentrations at which 1,10-p has been used as a transcriptional inhibitor (100 – 500 ug/mL) on p. 5.

3. The statement that toxicity of the inhibitor is not due to its effect on SUMOylation appears premature because it is also possible that an interference with rapid deSUMOylation is incompatible with long-term survival.

We have altered the text to indicate that the lethality is not due to direct or immediate effects of reduced SUMO conjugation levels (p. 10).

Response to Reviewers

4. The effects of the compound on mammalian cells are poorly characterized. A detailed study may be beyond the scope of this manuscript, but the differences between the SUMO isoforms are interesting and call for an explanation. Are there differences in the human conjugation system that explain this (could be tested in vitro) or is the kinetics of SUMO1 versus SUMO2/3 conjugates different in cells, perhaps due to the activity of isoform-specific isopeptidases? This would be relevant and interesting questions to ask.

Previously, two different people, using different techniques, had performed the analyses on the two human cell lines. To address this comment, we repeated the analysis using the same protocol for both cell lines in parallel, and performed the 1,10—p treatment using serum-free medium this time, as indicated on p. 19. As shown in new Fig. S1D, we now find a consistent effect for both SUMO isoforms and for both cell lines. The result is consistent with an inhibition of human sumoylation enzymes in vitro and with our observations of reduced sumoylation in yeast and in *Tetrahymena*.

As our study is focused largely on the yeast system, we agree that a detailed study of the effects on mammalian cells is beyond the scope of this manuscript. Undoubtedly, our findings will lead to further detailed studies in this area.

5. Given the role of Ulp1 as a processing enzyme for the SUMO precursor, is it possible that the synthetic lethal interaction between AA-Ubc9 and the Ulp1 mutant is due to an overall insufficient level of mature SUMO, i.e. would the expression of processed SUMO rescue the effect?

It is possible that a Ubc9-AA *ulp1-mt* strain would be deficient in mature SUMO peptide levels. However, the *ulp1-mt* strain itself is capable of a high level of SUMO conjugation, which might argue against that. Additionally, for the *ubc9-6* strain, which shows a more dramatic sumoylation defect than Ubc9-AA (see Fig. 3A in Moallem et al., 2023; PMID: 36720466), we were able to generate a viable *ubc9-6 ulp1-1* double mutant strain (Fig. 3E). Although curious and worth pursuing, the potential synthetic lethal interaction between Ubc9-AA and *ulp1-mt* is not key or significant to the present study.

Minor points:

1. The treatment times should be more precisely noted in some of the figures, e.g. Figure 1 and S1.

We have now indicated treatment times in the figure legends or figures themselves, throughout. All in vitro analyses (including that shown in Fig. S1A) were carried out for 1 h in the presence or absence of indicated concentrations of 1,10-p, as described in the Methods.

2. RT-PCR controls should be performed not only for thiolutin, but also for 1,10-phen.

As suggested, we have included 1,10-p-treated and thiolutin-treated samples in parallel in the RT-qPCR analysis shown in new Fig. 1E.

3. The switch between molar and mass-based concentrations for inhibitor concentration is annoying, as it makes comparisons difficult.

We used mass-based concentrations for in vivo experiments since that was the practice for treating yeast cultures with 1,10-p, but then switched to molarity for in vitro experiments to facilitate comparisons to other reaction components (i.e., Mg, ATP). To make it easier to compare

Response to Reviewers

concentrations in different experiments, we have now indicated the molar and mass-based concentrations where relevant directly on the figures or in the figure legends, including providing a conversion key at the bottom of Fig. 1C that we think will be helpful.

4. Fig. 3B: rapamycin treatment should also be performed on WT cells as a control.

This control was performed when the Ubc9-AA strain was first reported (Fig. 2A in Moallem et al., MCB, 2023; PMID: 36720466), but we have also now included it as new Fig. S1G.

Dear Dr. Rosonina

Thank you for the submission of your revised manuscript to EMBO reports. We have now received the full set of referee reports that is copied below.

As you will see, both referees are very positive about the study and support publication.

Browsing through the manuscript myself, I noticed a few editorial things that we need before we can proceed with the official acceptance of your study:

- Please update the 'Conflict of interest' paragraph to our new 'Disclosure and competing interests statement'. For more information see <https://www.embopress.org/page/journal/14693178/authorguide#conflictsofinterest>
- Please remove the Author Contributions from the manuscript file and make sure that the author contributions in our online submission system are correct and up-to-date. The information you specified in the system will be automatically retrieved and typeset into the article. You can enter additional information in the free text box provided, if you wish.
- Supplementary Figure 1 (and its legend) should be renamed to Appendix Figure S1 and the pdf should be called Appendix. The Appendix needs a title page and a table of content with page numbers (even if it is only one figure).
- Supplementary Table S1 provides quantitative data for Fig. 1E and Fig. 3F, which is also part of the source data you uploaded. Please reconsider whether you need the Supplementary table at all.
- Source data: please upload the source data as one separate folder per figure (currently you uploaded the data for all figures as one zip folder).
- The section "Supplemental items title" section should be removed from the manuscript.
- Reagents and Tools table: please add light grey filling to the subheadings, e.g. Antibodies, to clearly outline the subsections (according to the design of our template Reagent and Tools Table).
- The header of the methods section should be
Material and Methods
Subheading: Reagents and Tools table
Subheading: Methods and Protocols
- The Data Availability section should go before the Acknowledgments section
- Qiu et al 2021 is a preprint. Please cite it as (preprint: Qiu et al, 2021) in the text and add [PREPRINT] to the end of the citation in the reference list.
- Appendix Fig. S1: Westernblots seem to contain grid alignment which is possibly from the capturing software. Please check whether this grid alignment is also present on the original source files and not just a conversion error. Please provide the original unmodified source data for these Western blots.
- As a standard procedure, we edit the title and abstract of manuscripts to make them more accessible to a general readership. Please find the edited versions below my signature and please incorporate the suggestions into the manuscript word file, if you agree with these.
- Finally, EMBO Reports papers are accompanied online by A) a short (1-2 sentences) summary of the findings and their significance, B) 2-3 bullet points highlighting key results and C) a synopsis image that is 550x300-600 pixels large (width x height) in PNG for JPG format. You can either show a model or key data in the synopsis image. Please note that the size is rather small and that text needs to be readable at the final size. Please send us this information along with the revised manuscript.
- On a different note, I would like to alert you that EMBO Press offers a new format for a video-synopsis of work published with us, which essentially is a short, author-generated film explaining the core findings in hand drawings, and, as we believe, can be very useful to increase visibility of the work. This has proven to offer a nice opportunity for exposure i.p. for the first author(s) of the study.

Please see the following link for representative examples and their integration into the article web page:

<https://www.embopress.org/doi/full/10.15252/emboj.2019103932>

With kind regards,

Referee #1:

The authors have properly addressed the issues raised in my earlier review.

Specifically, they improved the quality of their study by demonstrating that 1,10-phenanthroline inhibited the activation step of the sumoylation pathway. Importantly, and consistent with this finding, the authors now show in the revised version of their manuscript that modification with different SUMO isoforms is affected similarly by the treatment of cells from two human cell lines with 1,10-phenanthroline. The differences between the different isoforms and cell lines in response to 1,10-phenanthroline treatment suggested in the earlier version of the manuscript were indeed not well characterised as rightly pointed out by the second reviewer.

Referee #2:

The authors have very thoroughly addressed my previous concerns and have strengthened the manuscript significantly. As expected, the revisions have not addressed the potential weakness inherent in the limited applicability of the inhibitor, but the authors have added important mechanistic information and clarified an apparent inconsistency between SUMO1 and SUMO2. Considering that they now show the mechanism by which their compound inhibits SUMOylation, I am actually very positive about this now well-controlled study and would be happy to see it published in EMBO Reports. I have no major concerns and found only two minor mistakes in the manuscript:

pg. 6: typo - 1,10-pheneanthrolin

pg. 23: Ref. Hendricks et al - issue and page numbers missing

The title should not exceed 100 characters incl. spaces and may not have punctuation. What about: 1,10-phenanthroline inhibits sumoylation and reveals that yeast SUMO modifications are highly transient

Abstract

The steady state levels of protein sumoylation depend on relative rates of conjugation and desumoylation. Whether SUMO modifications are generally long-lasting or short-lived is unknown. Here we show that treating budding yeast cultures with 1,10-phenanthroline abolishes most SUMO conjugations within one minute, without impacting ubiquitination, an analogous post-translational modification. 1,10-phenanthroline inhibits formation of the E1~SUMO thioester intermediate, demonstrating that it targets the first step in the sumoylation pathway. SUMO conjugations are retained after treatment with 1,10-phenanthroline in yeast that express a defective form of the desumoylase Ulp1, indicating that Ulp1 is responsible for eliminating existing SUMO modifications almost instantly, when de novo sumoylation is inhibited. This reveals that SUMO modifications are normally extremely transient because of continuous desumoylation by Ulp1. Supporting our findings, we demonstrate that sumoylation of two specific targets, Sko1 and Tfg1, virtually disappears within one minute of impairing de novo sumoylation. Altogether, we have identified an extremely rapid and potent inhibitor of sumoylation, and our work reveals that SUMO modifications are remarkably short-lived.

Response to Reviewers

Referee #1

The authors have properly addressed the issues raised in my earlier review. Specifically, they improved the quality of their study by demonstrating that 1,10-phenanthroline inhibited the activation step of the sumoylation pathway. Importantly, and consistent with this finding, the authors now show in the revised version of their manuscript that modification with different SUMO isoforms is affected similarly by the treatment of cells from two human cell lines with 1,10-phenanthroline. The differences between the different isoforms and cell lines in response to 1,10-phenanthroline treatment suggested in the earlier version of the manuscript were indeed not well characterised as rightly pointed out by the second reviewer.

Referee #2

The authors have very thoroughly addressed my previous concerns and have strengthened the manuscript significantly. As expected, the revisions have not addressed the potential weakness inherent in the limited applicability of the inhibitor, but the authors have added important mechanistic information and clarified an apparent inconsistency between SUMO1 and SUMO2.

Considering that they now show the mechanism by which their compound inhibits SUMOylation, I am actually very positive about this now well-controlled study and would be happy to see it published in EMBO Reports. I have no major concerns and found only two minor mistakes in the manuscript:

pg. 6: typo - 1,10-pheneanthrolin

pg. 23: Ref. Hendricks et al - issue and page numbers missing

We are delighted that both reviewers feel that their issues and concerns have been addressed and that our manuscript is now stronger and improved.

The two minor mistakes pointed out by Referee #2 have been corrected.

We thank both referees for their very useful suggestions and all their work on this manuscript.

Dr. Emanuel Rosonina
York University
Biology
4700 Keele Street
329D Life Sciences Building
Toronto, ON Canada M3J 1P3
Canada

Dear Dr. Rosonina,

I am very pleased to accept your manuscript for publication in the next available issue of EMBO reports. Thank you for your contribution to our journal.

Yours sincerely,
